# Reward-Consistent Dynamics Models are Strongly Generalizable for Offline Reinforcement Learning

**Fan-Ming Luo, Tian Xu, Xingchen Cao & Yang Yu**[*]
National Key Laboratory for Novel Software Technology, Nanjing University, China
School of Artificial Intelligence, Nanjing University, China
Polixir.ai
`{luofm,xut,caoxc,yuy}@lamda.nju.edu.cn`

## Abstract

Learning a precise dynamics model can be crucial for offline reinforcement learning, which, unfortunately, has been found to be quite challenging. Dynamics models that are learned by fitting historical transitions often struggle to generalize to unseen transitions. In this study, we identify a hidden but pivotal factor termed *dynamics reward* that remains consistent across transitions, offering a pathway to better generalization. Therefore, we propose the idea of reward-consistent dynamics models: any trajectory generated by the dynamics model should maximize the dynamics reward derived from the data. We implement this idea as the MOREC (Model-based Offline reinforcement learning with Reward Consistency) method, which can be seamlessly integrated into previous offline model-based reinforcement learning (MBRL) methods. MOREC learns a generalizable dynamics reward function from offline data, which is subsequently employed as a transition filter in any offline MBRL method: when generating transitions, the dynamics model generates a batch of transitions and selects the one with the highest dynamics reward value. On a synthetic task, we visualize that MOREC has a strong generalization ability and can surprisingly recover some distant unseen transitions. On 21 offline tasks in D4RL and NeoRL benchmarks, MOREC improves the previous state-of-the-art performance by a significant margin, i.e., 4.6% on D4RL tasks and 25.9% on NeoRL tasks. Notably, MOREC is the first method that can achieve above 95% online RL performance in 6 out of 12 D4RL tasks and 3 out of 9 NeoRL tasks. Code is available at `https://github.com/polixir/morec`.

## 1 Introduction

Model-based reinforcement learning (MBRL) approaches encompass techniques that harness either an established environment model or one learned to approximate the environment, thereby addressing RL challenges (Sutton & Barto, 1998; Luo et al., 2023; Moerland et al., 2023). These models are primarily employed to forecast forthcoming states arising from the execution of actions within the environment. By leveraging models, it becomes possible to evaluate action sequences or policies through simulation, circumventing the need for direct interactions with the actual environment and substantially curtailing sampling expenses. As a result, model-based methods empower offline RL scenarios where agents exclusively operate with a dataset sampled from the environment, devoid of direct access to the environment itself, thus enabling the adoption of a range of efficient methodologies (Yu et al., 2020; 2021; Rigter et al., 2022; Sun et al., 2023).

One can conduct numerous trial searches within an ideal model until the optimal policies for a task are discovered. However, it is incredibly challenging to learn a uniformly accurate dynamics model solely through initial supervised learning on the offline dataset. Without additional guidance on the models, model errors will inevitably occur. These errors become particularly pronounced in state-action pairs that fall outside the distribution of the offline data. Such instances may arise when

---

[*]Yang Yu is the corresponding author.

dealing with long forecasting horizons or when the learning policies deviate significantly from the behavior policy of the offline data. These model errors have the potential to be erroneously exploited by algorithms, consequently yielding policies that exhibit significant underperformance.

Research on mitigating model error focuses on two main branches: enhancing policy learning to avoid model inaccuracies and improving model learning techniques. MOPO (Yu et al., 2020) introduced penalizing uncertain rewards and shortening model simulation horizons to reduce policy deviations and minimize out-of-distribution states visits. Subsequent research introduced more sophisticated conservative strategies, such as MOReL's (Kidambi et al., 2020) early stopping and RAMBO's (Rigter et al., 2022) pessimistic models, while MOBILE (Sun et al., 2023) enhanced uncertainty estimation for broader policy exploration. On the other hand, non-conservative methods like MAPLE (Chen et al., 2021) aimed at improving generalization through context-aware meta-policy learning, though their success is ultimately limited by the learned model's capabilities.

For better model learning, VirtualTaobao (Shi et al., 2019) led with adversarial model learning for improved accuracy, a technique further applied in complex industrial contexts (Shang et al., 2021). This adversarial approach has been effective in addressing compounding errors (Xu et al., 2022a;b; 2023), enabling longer trajectory simulations. Moreover, adversarial counterfactual model learning (Chen et al., 2023) proved adept at learning causal transitions challenging for straightforward supervised learning methods.

This paper firstly focuses on learning better models. To enhance the generalization ability of dynamics models, our goal is to identify the invariance underlying the data. We identify a hidden but pivotal factor named **dynamics reward**, which represents the inherent driving forces of the environment dynamics. For instance, a naive dynamics reward is a function that assigns 1 to true transitions and 0 to false transitions. We can conceptualize the dynamics model as an *agent* which maximizes the dynamics reward (Xu et al., 2022a). We then propose the idea of reward-consistent dynamics models: any trajectory generated by the dynamics model should maximize the dynamics reward derived from the data. To implement this idea, we learn a dynamics reward function through an inverse reinforcement learning (IRL) method from the offline data. We finally incorporate the dynamics reward into the policy learning stage by modifying the model-rollout procedure. Specifically, we (i) regulate the model's output to generate rollouts associated with higher dynamics rewards; (ii) terminate the rollout upon encountering states with low dynamics rewards. This method can be integrated into most prior offline MBRL methods, resulting in a novel one named MOREC (Model-based Offline policy optimization with REward Consistency).

In experiments, we first evaluate MOREC on a synthetic refrigerator temperature-control task. We visualize that the learned dynamics reward aligns closely with the accuracy of transitions, even in OOD regions, demonstrating its superior generalization ability. Furthermore, we demonstrate that the utilization of such a generalizable dynamics reward facilitates the generation of high-fidelity model rollouts, leading to a substantial enhancement in policy performance. Additionally, we evaluate MOREC on 21 typical tasks from two offline benchmarks, D4RL (Fu et al., 2020) and NeoRL (Qin et al., 2022). The empirical results show MOREC outperforms prior state-of-the-art (SOTA) methods on 18 out of the 21 tasks. Notably, in the more difficult NeoRL benchmark, MOREC achieves a remarkable $25.9\%$ average performance improvement over prior SOTAs and solves 3 tasks for the first time (0 previously). In-depth analysis shows a strong correlation between the learned dynamics rewards and model prediction errors. Guided by such an instructive dynamics reward, MOREC shows a significant reduction of the model prediction errors even when a long rollout horizon is adopted.

## 2 PRELIMINARIES AND RELATED WORK

**Reinforcement Learning (RL).** In RL, we consider the Markov decision process (MDP) (Sutton & Barto, 1998), described by a tuple $\langle \mathcal{S}, \mathcal{A}, P^\star, r^{\text{task}}, \gamma, \rho_0 \rangle$, where $\mathcal{S}$ is the state space, $\mathcal{A}$ is the action space, $P^\star : \mathcal{S} \times \mathcal{A} \to \Delta(\mathcal{S})$ is the true dynamics, $r^{\text{task}} : \mathcal{S} \times \mathcal{A} \to \mathbb{R}$ is the task reward function, $\gamma \in (0, 1)$ is the discount factor, and $\rho_0$ is the initial state distribution. The value function $V^\pi(s) = \mathbb{E}_\pi \left[ \sum_{i=0}^\infty \gamma^i r^{\text{task}}(s_{t+i}, a_{t+i}) \mid s_t = s \right]$ is the expected cumulative rewards when starting at state $s_t = s$ and following $\pi$. Similarly, we can define the action-value function as $Q^\pi(s, a) = \mathbb{E}_\pi \left[ \sum_{i=0}^\infty \gamma^i r^{\text{task}}(s_{t+i}, a_{t+i}) \mid s_t = s, a_t = a \right]$. The objective of RL is to find a policy $\pi$ that maximizes the expected value under the initial state distribution, i.e., $\max_\pi \mathbb{E}_{s_0 \sim \rho_0(\cdot)} \left[ V^\pi(s_0) \right]$.

Figure 1: Framework of MOREC. Key distinctions between MOREC and the prior offline MBRL are emphasized in red. The main modifications arise in the model learning and model simulation stages.

**Offline RL.** Offline RL (Levine et al., 2020) emphasizes learning from an offline dataset without additional interaction with the environment. Here the offline dataset $\mathcal{D} = \{(s, a, r^{\text{task}}, s')\}$ consists of transitions from trajectories collected by a behavior policy $\pi_\beta$. A primary obstacle faced in this domain arises from discrepancies between the offline data and the behavior policy, which leads to extrapolation errors (Kumar et al., 2019). To mitigate these errors, model-free offline RL methods have integrated conservatism, either by modulating the policy or the Q-function of online RL algorithms (Fujimoto et al., 2019; Bai et al., 2022).

**Offline MBRL.** Building upon the foundational concepts of offline RL, offline MBRL introduces an advanced strategy by leveraging a dynamics model constructed from offline data. Offline MBRL methods typically possess two stages: (i) learn a model of the environment from the offline data $\mathcal{D}$; (ii) learn a policy from the model and $\mathcal{D}$. Let the model parameterized by $\theta$ be $P_\theta(s_{t+1}|s_t, a_t)$. It will be trained by log-likelihood maximization:

$$\max_\theta \ \mathcal{L}_M(\theta) := \mathbb{E}_{(s,a,s') \sim \mathcal{D}} \left[ \log(P_\theta(s'|s, a)) \right]. \tag{1}$$

Without loss of generality, we assume that the task reward function $r^{\text{task}}$ is known since we can simply incorporate $r^{\text{task}}$ into the dynamics model if it unknown. The strategy of offline MBRL allows for the generation of synthetic data, potentially enhancing adaptability to out-of-distribution states (Yu et al., 2020; Kidambi et al., 2020). However, the learned dynamics model also inevitably suffers from errors for the limited experience dataset (Xu et al., 2022a). To mitigate the challenges stemming from model inaccuracies, several approaches integrate conservatism through methods like uncertainty estimation to guide the behavior policy closer to the available dataset (Yu et al., 2020). The most recent works focus on designing better conservative strategies (Kidambi et al., 2020; Sun et al., 2023; Rigter et al., 2022) to unleash the full potential of the model. However, these methods are still limited by the static capabilities of the learned model itself. In this paper, we present a novel approach to enhance the fidelities of model rollouts, through guidance of reward signals. This method can not only seamlessly integrate seamlessly but also augment previous offline MBRL techniques.

**Inverse Reinforcement Learning (IRL).** IRL (Ng & Russell, 2000; Ni et al., 2020) aims to recover a reward function from expert demonstrations. A main class of algorithms (Abbeel & Ng, 2004; Ho & Ermon, 2016; Swamy et al., 2021) in IRL infer a reward function by maximizing the value gaps between the expert policy and the other policies, as the expert policy should perform well under the desired reward function. It is worth noting that previous IRL methods recover a reward function for a policy. In this paper, we apply IRL to learn a reward function for the dynamics model.

## 3 METHOD

In this section, we will delve into the details of how we learn the dynamics reward function and how we leverage this reward function to facilitate generating high-fidelity model rollouts. We begin with an overview of our method in Section 3.1, followed by an explanation of dynamics reward learning in Section 3.2. At last, we present the complete process of MOREC in Section 3.3.

### 3.1 OVERVIEW OF MOREC

We aim to improve the fidelity of model rollouts along with the policy improvement. To accomplish this, we must uncover the underlying invariance across different data instances. We have identified a crucial hidden factor called dynamics reward, which represents the intrinsic driving forces of the environment dynamics. By considering the dynamics system as an agent (Xu et al., 2022a), we can

assume that the dynamics system was also learned by maximizing the dynamics reward. Therefore, regardless of the policy used to generate interaction data, the data should exhibit consistent dynamics reward.

Based on this idea, we develop a new offline MBRL framework MOREC, which is illustrated in Fig. 1. As a general framework, MOREC can be applied to the most existing model-based offline RL methods such as MOPO (Yu et al., 2020), MOREL (Kidambi et al., 2020), and MOBILE (Sun et al., 2023). Compared with the framework of prior model-based offline RL methods, MOREC incorporates two significant algorithmic designs, which are marked by red color in Fig. 1. First, we apply IRL to infer a dynamics reward function tailored for the dynamics model. We elaborate on the training process of the dynamics reward in Section 3.2. The second algorithmic design is located in the model simulation stage. In particular, we utilize the learned dynamics reward to help generate high-fidelity model rollouts. The detailed generation procedure is explained in Section 3.3.

## 3.2 DYNAMICS REWARD LEARNING

In this part, we present how we learn a reward function specially designed for dynamics models. Our method is motivated by the new perspective of dynamics models. That is, we can view the dynamics model as an "agent" with the new augmented state space $\mathcal{S} \times \mathcal{A}$ and the new action space $\mathcal{S}$. Accordingly, the behavior policy $\pi_\beta$ is regarded as the "dynamics model". Besides, there is a dynamics reward function $r^D : \mathcal{S} \times \mathcal{A} \times \mathcal{S} \to \mathbb{R}$ that can judge the performance of a dynamics model. For instance, a naive dynamics reward is a function that assigns $1$ on true transitions and $0$ on false transitions. Under the dynamics reward, the true dynamics is a good "policy" that achieves the maximum dynamics reward. From this perspective, we can formulate the problem of dynamics reward learning as an IRL problem, which aims to infer a reward function from expert demonstrations collected by the expert policy. In particular, the true dynamics is viewed as the expert policy and the offline dataset collected in the true dynamics is viewed as the expert demonstrations in IRL. Our target is to learn a reward function that can explain the true dynamics (i.e., the expert policy).

Based on the above IRL formulation, we leverage the advancements in IRL to help infer the dynamics reward. In particular, inspired by the well-known IRL method GAIL (Ho & Ermon, 2016), we consider the following dynamics reward learning objective.

$$\ell(D, P) = \mathbb{E}_{(s,a,s')\sim\mathcal{D}}\left[\log(D(s, a, s'))\right] + \mathbb{E}_{(s,a)\sim\mathcal{D},s'\sim P(\cdot|s,a)}\left[\log(1 - D(s, a, s'))\right]. \quad (2)$$

With a slight abuse of notation, we use $\mathcal{D}$ to denote both the state-action and state-action-next-state distributions from the offline dataset. Here $P : \mathcal{S} \times \mathcal{A} \to \Delta(\mathcal{S})$ is an arbitrary dynamics model and $D : \mathcal{S} \times \mathcal{A} \times \mathcal{S} \to (0, 1)$ is a discriminator, which is used to induce the dynamics reward. More concretely, when we consider the dynamics reward of $r^D(s, a, s') = -\log(1 - D(s, a, s'))$ (Ho & Ermon, 2016), $\ell(D, P)$ can be roughly interpreted as the difference between rewards on the true dynamics $P^\star$ and on the dynamics model $P$. Notice that a desired dynamics reward should induce a large reward difference $\ell(D, P)$ since the true dynamics achieves the maximal reward on such a dynamics reward function.

Following the maximum margin IRL method (Abbeel & Ng, 2004; Ratliff et al., 2006; Ho & Ermon, 2016), we further define the margin function $f(D) := \min_P \ell(D, P)$ and consider maximizing such a margin function.

$$\max_D f(D) = \max_D \min_P \mathbb{E}_{(s,a,s')\sim\mathcal{D}}\left[\log(D(s, a, s'))\right] + \mathbb{E}_{(s,a)\sim\mathcal{D},s'\sim P(\cdot|s,a)}\left[\log(1 - D(s, a, s'))\right].$$

In other words, we aim to find a dynamics reward that can maximize the reward difference with an *adversarial dynamics model $P$*. To optimize this objective, we apply the gradient-descent-ascent method (Lin et al., 2020) which alternates between updating the dynamics reward and adversarial dynamics model. The inner problem corresponds to a special case of RL where the horizon equals one, and thus we can utilize the policy gradient method (Sutton et al., 1999) to update the adversarial dynamics model. The outer problem is exactly the binary classification problem by considering the transitions in $\mathcal{D}$ as positive samples and the transitions generated by $P$ as negative samples. The detailed procedure is outlined in Algorithm 2 in Appendix B.1. Note that GAN (Goodfellow et al., 2014) can also be applied in solving the above problem. Here we choose the policy gradient method because it is also applicable in the general RL set-up, which is considered in most IRL works (Abbeel & Ng, 2004; Ratliff et al., 2006; Ziebart et al., 2008; Ho & Ermon, 2016). Connections between MOREC and adversarial model learning are discussed in Appendix C.

---

**Algorithm 1:** MOREC-MOPO

---

**Input:** Offline data $\mathcal{D}$; initialized policy $\pi_\phi$ and critic $Q_\psi$; uncertainty penalty coefficient $\beta$;

1 Invoke Algorithm 2 to learn the dynamics reward $r^D$ ;
2 Learn a model ensemble $\mathcal{P} = \{p_{\theta_i}(s'|s, a), \forall i \in [M]\}$ via Eq.(1) ;
3 Initialize an empty buffer $\mathcal{B} \leftarrow \varnothing$;
4 **for** $i = 1, \ldots, N_{iter}$ **do**
5    **for** $j = 1, \ldots, N_{rollout}$ **do**
6       Sample a state $s_t$ from $\mathcal{D}$ uniformly ;
7       Generate a rollout $l = \{s_t, a_t, r_t^{\mathrm{raw}}, ...\}$ following Eq.(4) with the dynamics reward $r^D$;
8       Minus the rewards with uncertainty penalties $r_t \leftarrow r_t^{\mathrm{raw}} - \beta\mathcal{U}(\mathcal{P}, s_t, a_t, s_{t+1}), \forall r_t^{\mathrm{raw}} \in l$;
9       Insert $\{s_t, a_t, r_t, s_{t+1}, \ldots\}$ to $\mathcal{B}$;
10    Draw samples from $\mathcal{D} \cup \mathcal{B}$ and apply SAC to update $\pi_\phi$ and $Q_\psi$;

---

We theoretically justify the dynamics reward learning algorithm by providing a convergence analysis.

**Proposition 1.** *Consider Algorithm 2. Assume that the gradient norm is bounded, i.e., $\forall t \in [T], \|\nabla_P \ell(D_t, P_t)\|_2 \leq G^P, \|\nabla_D \ell(D_t, P_t)\|_2 \leq G^D$. If we take the step size of $\{\eta_t^P = \sqrt{|\mathcal{S}||\mathcal{A}|}/(G^P t), \eta_t^D = \sqrt{|\mathcal{S}|^2|\mathcal{A}|}/(G^D t)\}_{t=1}^T$, then we have*

$$f\left(\frac{1}{T}\sum_{t=1}^T D_t\right) \geq \max_D f(D) - 3G^D\sqrt{\frac{|\mathcal{S}|^2|\mathcal{A}|}{T}} - 3G^P\sqrt{\frac{|\mathcal{S}||\mathcal{A}|}{T}}.$$

Please refer to Appendix B.2 for thorough proof. Proposition 1 indicates that the proposed dynamics reward learning algorithm can converge to the global optimum with a rate of $\mathcal{O}(1/\sqrt{T})$. Besides, Proposition 1 addresses the tabular setting with tabular discriminator and dynamics models. This can be generalized to the linear function approximation setting (Ayoub et al., 2020; Cai et al., 2020), as detailed in Appendix B.3. Finally, we remark that Proposition 1 shows an average-iterate convergence (Nesterov, 2003). To consist with the theory, in practice, we maintain an ensemble of discriminators in the training procedure and output the dynamics reward of

$$r^D(s, a, s') = -\log\left(1 - \frac{1}{T}\sum_{t=1}^T D_t(s, a, s')\right), \quad \forall (s, a, s') \in \mathcal{S} \times \mathcal{A} \times \mathcal{S}. \tag{3}$$

### 3.3 MODEL-BASED OFFLINE POLICY OPTIMIZATION WITH REWARD CONSISTENY

In this part, we elaborate on how MOREC incorporates the dynamics reward for guiding the dynamics model in generating model rollouts. The initial learning of dynamics models in MOREC is the same as that in the prior offline MBRL framework. In particular, we learn a model ensemble, denoted as $\{P_{\theta_m}(s'|s, a), \forall m \in [M]\}$, where $\theta_m$ represents the parameter of the $m$-th model, $M$ symbolizes the ensemble size and $[M] := \{1, 2, \cdots, M\}$. Each model in this ensemble is learned through supervised learning using Eq.(1).

When using the learned dynamics models to generate samples, MOREC incorporates the dynamics reward to select high-fidelity transitions from all sampled candidates, which is the key difference from other existing model-based offline RL methods. More specifically, at time step $t$, MOREC begins by sampling from each transition distribution $P_{\theta_i}(\cdot, |s_t, a_t)$ $N$ times to obtain $MN$ candidate of next states. The selections are then made based on a probability distribution induced by the softmax function, which favors transitions with higher dynamics rewards. The softmax distribution also incorporates uncertainty into the model rollout process, which has been previously found to be helpful for policy generalization (Chua et al., 2018). The detailed rollout generation process is:

$$a_t \sim \pi_\phi(\cdot|s_t); \ \ s_{t+1}^{m,n} \sim P_{\theta_m}(\cdot|s_t, a_t), \ \forall m \in [M], n \in [N]; \ \ s_{t+1} = s_{t+1}^{m^\star, n^\star};$$

$$\text{where } m^\star, n^\star \sim P(\cdot), \ P(m, n) = \frac{\exp(r^D(s_t, a_t, s_{t+1}^{m,n})/\kappa)}{\sum_{m'\in[M]}\sum_{n'\in[N]}\exp(r^D(s_t, a_t, s_{t+1}^{m',n'})/\kappa)}. \tag{4}$$

Here $\pi_\phi$ is the current learning policy with parameter $\phi$ and $\kappa$ is the temperature coefficient of the softmax function. The above process is repeated for $H_{\mathrm{rollout}}$ times to generate a trajectory with the

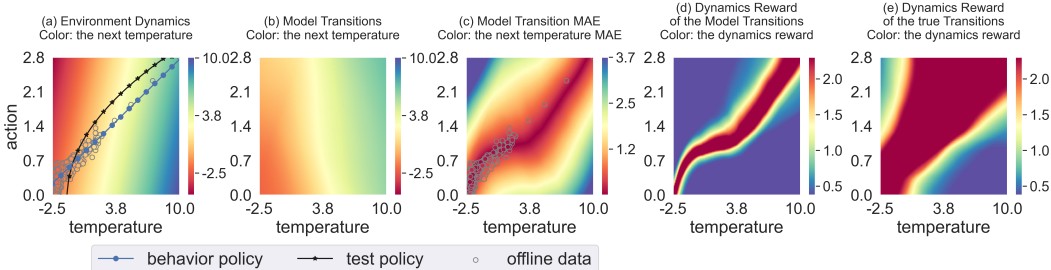

Figure 2: Illustration of the (a) environment dynamics, (b) dynamics model, (c) model transition MAE, (d) dynamics reward of the model, and (e) dynamics reward of the true transitions in the refrigerator temperature-control task. Panel (a) visualizes the relationship between temperature, action, and the next temperature, mapped to the $x$, $y$, and color axes, respectively. A point at coordinate $(x, y)$, colored by value $c$, indicates that applying action $y$ at temperature $x$ results in a next temperature of $c$. For panels (b)-(e), the meaning of the color axes is clarified within the respective figure titles.

maximal length $H_{\text{rollout}}$. Besides, in order to manage cases where the chosen transitions deviate significantly from the true one, leading to an extremely low dynamics reward, early rollout termination is also implemented when $r^D(s_t, a_t, s_{t+1}) < r^D_{\min}$, where $r^D_{\min}$ is a preset hyperparameter. With the above transition filtering technique, we can leverage the dynamics reward to generate high-fidelity transitions, which benefit the subsequent policy learning phase.

As for the policy learning phase, the developed framework MOREC can incorporate the policy learning techniques in the most existing offline MBRL methods. In Algorithm 1, we provide an instantiation MOREC-MOPO, which utilizes MOPO (Yu et al., 2020) as the policy update component. The uncertainty estimation in line 8 is elaborated in Appendix D.2. In experiments, we also apply MOREC on the SOTA model-based offline RL method MOBILE (Sun et al., 2023).

## 4 EXPERIMENTS

In this section, we conduct a series of experiments designed to answer the following questions:
**Q1:** What is the appearance of the recovered dynamics reward? (Fig. 2, 5)
**Q2:** Can the recovered dynamics reward enhance the accuracy of rollouts? (Fig. 3, 4, 6)
**Q3:** Can MOREC facilitate learning policies with superior performance? (Table 1, 2, 3)

### 4.1 EXPERIMENTAL SETUP

In our experiments, we consider a synthetic refrigerator temperature-control task, 12 locomotion tasks as part of the D4RL benchmark (Fu et al., 2020), and 9 locomotion tasks included in the NeoRL benchmark (Qin et al., 2022). Comprehensive experimental details are provided in Appendix E.

### 4.2 EVALUATION ON A SYNTHETIC TASK

We first consider a synthetic refrigerator temperature-control task where we can delve deeply into the performance of MOREC through visualizations. In this task, an agent controls the compressor's power in a refrigerator with the primary aim of maintaining a preset target temperature. The agent's observations are the present temperature, with the action corresponding to the normalized power of the compressor. We depict the dynamics of this system in Fig. 2 (a), which maps temperature, action, and the next temperature to the $x$, $y$, and color axes, respectively. We then use a linear behavior policy to collect the offline dataset (also shown in Fig. 2 (a)) from the environment.

We train a dynamics model with the offline data which is shown in Fig. 2 (b). To compare the dynamics model and the true dynamics more clearly, we showcase the mean absolute error (MAE) between the next temperatures output by the true dynamics and the dynamics model in Fig. 2 (c). A comparative analysis of Fig. 2 (a), (b), and (c) suggests the model can align closely with the true dynamics where the offline data is dense, but deviate in data-scarce areas. Notably, the low-error area in Fig. 2 (c), denoted in red, mainly corresponds to the regions visited by the behavior policy.

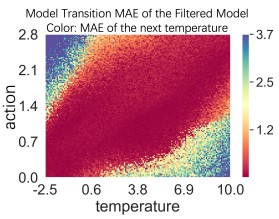

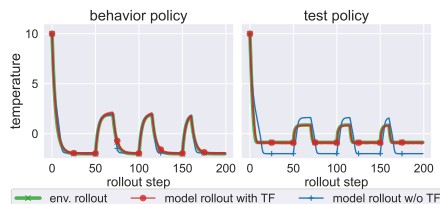

Table 1: Returns of the behavior policy and the policies learned with or without transition filtering in the true environment.

Figure 3: MAE of the transitions filtered by the dynamics reward.

Figure 4: 200-step rollouts in model or environment with the behavior policy or test policy (TF: transition filtering).

| Method | Return |
|---|---|
| with transition filtering | $-78$ |
| w/o transition filtering | $-640$ |
| behavior policy | $-270$ |

To answer **Q1**, we further train a dynamics reward according to Algorithm 2. We depict the inferred dynamics rewards of the model transitions and true transitions in Fig. 2 (d) and (e), respectively. A noteworthy observation is the strong correlation between the inferred dynamics rewards and the MAE of the next temperatures, as shown in Fig. 2 (c) and (d). The dynamics reward aptly allocates higher values to model transitions corresponding to a lower MAE. Moreover, the dynamics rewards of the true transitions (Fig. 2 (e)) are higher than the rewards of the model (Fig. 2 (d)), in almost all of the region. This shows that the dynamics reward model can serve as a scoring function to evaluate the truthfulness of transitions, even in the OOD region.

To answer **Q2**, we use the proposed transition filtering technique detailed in Eq.(4) to improve the transitions of the dynamics model. We present the MAE of the filtered model in Fig. 3. Compared with Fig. 2 (c), Fig. 3 shows a expanded low-error region. This clearly demonstrates the superior performance of utilizing the dynamics reward for transition filtering. A direct comparison between the figures is shown in Fig. 7 in Appendix F.1. Subsequently, we compare the accuracy of model rollouts generated with and without the transition filtering technique. Here we consider two rollout policies: the behavior policy and another different test policy which is illustrated in Fig. 2 (a). Fig. 4 shows the true environment rollout and model rollouts. First, we focus on the behavior policy case, which is shown in the left sub-figure in Fig. 4. As the dynamics model is trained on the offline dataset collected by the behavior policy, it can give accurate next-state predictions on in-distribution state-action pairs visited by the behavior policy. Thus, both model rollouts with and without transition filtering can perfectly replicate the true environment rollout.

However, in model-based offline RL, our primary concern is the accuracy of model rollouts induced by learning policies, which could deviate from the behavior policy and visit out-of-distribution (OOD) state-action pairs. Thus we turn to the test policy case shown in the right sub-figure in Fig. 4. As the dynamics model cannot give accurate predictions in a single attempt on OOD state-action pairs visited by the test policy, the model rollout generated without the transition filtering technique deviates from the true one significantly. Nevertheless, when equipped with the transition filtering technique, the generated model rollout still perfectly matches the true rollout. These results clearly demonstrate the effectiveness of utilizing the dynamics reward to generate high-fidelity model rollouts on OOD regions. This further suggests that the learned dynamics reward could exhibit a superior generalization ability than the dynamics model.

To answer **Q3**, we train a policy that aims to control the temperature to $-1°C$ with the reward function $r(\text{temperature}_t) = -|\text{temperature}_t + 1|$. In particular, we apply PPO (Schulman et al., 2017) to learn policies from model rollouts generated with and without the proposed transition filtering technique. The detailed returns of the learned policies are reported in Table 1. We see that the policy learned with transition filtering outperforms that learned without transition filtering by a wide margin. These observations validate that utilizing the dynamics reward can substantially boost the performance of policy learning. The same conclusion can also be obtained by comparing the rollouts of the learned policies in the *true* environment in Fig. 8 in Appendix F.1.

### 4.3 EVALUATION ON THE D4RL AND NEORL BENCHMARK

To assess the applicability of MOREC in more complex tasks, we validate it on 21 robot locomotion control tasks from the D4RL (Fu et al., 2020) and NeoRL (Qin et al., 2022) benchmarks. Notice that the NeoRL benchmark is more challenging than the D4RL benchmark as the offline datasets in

Table 2: Normalized average returns in 12 D4RL tasks, averaged over 5 seeds. *Solved tasks* denotes the number of the tasks whose scores $\geq 95.0$. The previous SOTA methods are underlined.

| Task Name | CQL | TD3+BC | EDAC | MOPO | COMBO | TT | RAMBO | MOBILE | MOREC-MOPO | MOREC-MOBILE |
|---|---|---|---|---|---|---|---|---|---|---|
| hfctah-rnd | 31.3 | 11.0 | 28.4 | 38.5 | 38.8 | 6.1 | 39.5 | 39.3 | $51.6 \pm 0.5$ | $\mathbf{53.2 \pm 1.4}$ |
| hopper-rnd | 5.3 | 8.5 | 25.3 | 31.7 | 17.9 | 6.9 | 25.4 | 31.9 | $\mathbf{32.1 \pm 0.1}$ | $26.6 \pm 4.0$ |
| walker-rnd | 5.4 | 1.6 | 16.6 | 7.4 | 7.0 | 5.9 | 0.0 | 17.9 | $\mathbf{23.5 \pm 0.7}$ | $22.8 \pm 0.9$ |
| hfctah-med | 46.9 | 48.3 | 65.9 | 73.0 | 54.2 | 46.9 | 77.9 | 74.6 | $\mathbf{82.3 \pm 1.1}$ | $82.1 \pm 1.2$ |
| hopper-med | 61.9 | 59.3 | 101.6 | 62.8 | 97.2 | 67.4 | 87.0 | 106.6 | $107.0 \pm 0.3$ | $\mathbf{108.0 \pm 0.5}$ |
| walker-med | 79.5 | 83.7 | 92.5 | 84.1 | 81.9 | 81.3 | 84.9 | 87.7 | $\mathbf{89.9 \pm 0.7}$ | $85.8 \pm 0.5$ |
| hfctah-med-rep | 45.3 | 44.6 | 61.3 | 72.1 | 55.1 | 44.1 | 68.7 | 71.7 | $\mathbf{76.5 \pm 1.2}$ | $76.4 \pm 0.6$ |
| hopper-med-rep | 86.3 | 60.9 | 101.0 | 103.5 | 89.5 | 99.4 | 99.5 | 103.9 | $105.1 \pm 0.4$ | $\mathbf{105.5 \pm 1.6}$ |
| walker-med-rep | 76.8 | 81.8 | 87.1 | 85.6 | 56.0 | 82.6 | 89.2 | 89.9 | $95.5 \pm 2.0$ | $\mathbf{95.8 \pm 1.8}$ |
| hfctah-med-exp | 95.0 | 90.7 | 106.3 | 90.8 | 90.0 | 95.0 | 95.4 | 108.2 | $112.1 \pm 1.8$ | $\mathbf{110.9 \pm 0.56}$ |
| hopper-med-exp | 96.9 | 98.0 | 110.7 | 81.6 | 111.1 | 110.0 | 88.2 | 112.6 | $\mathbf{113.3 \pm 0.2}$ | $111.5 \pm 0.4$ |
| walker-med-exp | 109.1 | 110.1 | 114.7 | 112.9 | 103.3 | 101.9 | 56.7 | **115.2** | $115.8 \pm 0.8$ | $115.5 \pm 0.9$ |
| Average | 61.6 | 58.2 | 76.0 | 70.3 | 66.8 | 62.3 | 67.7 | 80.0 | $\mathbf{83.7}$ | 82.8 |
| Solved tasks | 3/12 | 2/12 | 5/12 | 2/12 | 3/12 | 4/12 | 2/12 | 5/12 | $\mathbf{6/12}$ | $\mathbf{6/12}$ |

Table 3: Normalized average returns in 9 NeoRL tasks, averaged over 5 seeds. *Solved tasks* denotes the number of the tasks whose scores $\geq 95.0$. The previous SOTA methods are underlined.

| Task Name | BC | CQL | TD3+BC | EDAC | MOPO | MOBILE | MOREC-MOPO | MOREC-MOBILE |
|---|---|---|---|---|---|---|---|---|
| HalfCheetah-L | 29.1 | 38.2 | 30.0 | 31.3 | 40.1 | **54.7** | $53.5 \pm 0.6$ | $51.2 \pm 0.99$ |
| Hopper-L | 15.1 | 16.0 | 15.8 | 18.3 | 6.2 | 17.4 | $25.4 \pm 1.3$ | $\mathbf{26.8 \pm 1.4}$ |
| Walker2d-L | 28.5 | 44.7 | 43.0 | 40.2 | 11.6 | 37.6 | $\mathbf{65.0 \pm 1.3}$ | $56.8 \pm 1.9$ |
| HalfCheetah-M | 49.0 | 54.6 | 52.3 | 54.9 | 62.3 | 77.8 | $84.1 \pm 0.5$ | $\mathbf{86.0 \pm 1.3}$ |
| Hopper-M | 51.3 | 64.5 | 70.3 | 44.9 | 1.0 | 51.1 | $83.5 \pm 3.8$ | $\mathbf{109.2 \pm 0.6}$ |
| Walker2d-M | 48.7 | 57.3 | 58.5 | 57.6 | 39.9 | 62.2 | $\mathbf{76.6 \pm 1.7}$ | $71.1 \pm 0.6$ |
| HalfCheetah-H | 71.3 | 77.4 | 75.3 | 81.4 | 65.9 | 83.0 | $90.3 \pm 2.0$ | $\mathbf{98.5 \pm 0.8}$ |
| Hopper-H | 43.1 | 76.6 | 75.3 | 52.5 | 11.5 | 87.8 | $72.8 \pm 3.8$ | $\mathbf{108.5 \pm 0.3}$ |
| Walker2d-H | 72.6 | 75.3 | 69.6 | 75.5 | 18.0 | 74.9 | $\mathbf{83.0 \pm 1.4}$ | $79.3 \pm 0.6$ |
| Average | 45.4 | 56.1 | 54.5 | 50.7 | 28.5 | 60.7 | 70.3 | $\mathbf{76.4}$ |
| Solved tasks | 0/9 | 0/9 | 0/9 | 0/9 | 0/9 | 0/9 | 0/9 | $\mathbf{3/9}$ |

NeoRL are collected from more narrow distributions (Qin et al., 2022).

**Overall Performance.** The normalized average returns are reported in Tables 2 and 3. We observe that MOREC-MOPO and MOREC-MOBILE outperform prior offline RL methods on 18 and 17 of total 21 tasks, respectively. On both the D4RL and NeoRL benchmarks, MOREC-MOPO and MOREC-MOBILE respectively achieve substantial improvements over MOPO and MOBILE, which clearly demonstrate the benefit from the dynamics reward on policy learning and thus effectively respond to **Q3**. In the more challenging NeoRL benchmark, MOREC offers a more significant improvement over the existing offline RL methods. In particular, MOREC-MOBILE outperforms the SOTA method MOBILE in terms of the average return by a wide margin of 15.7, representing an approximate improvement of 25.9%. Furthermore, we also report the number of solved tasks which refer to tasks with normalized returns exceeding 95. In NeoRL, MOREC-MOBILE is the first method that can solve three tasks while previous methods could not solve any of them. We present the learning curves and the results on `Adroit` tasks in Appendix F.6 and F.8, respectively.

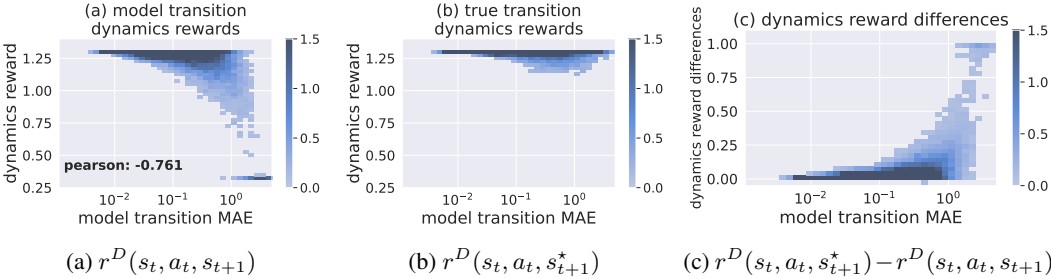

(a) $r^D(s_t, a_t, s_{t+1})$      (b) $r^D(s_t, a_t, s_{t+1}^{\star})$      (c) $r^D(s_t, a_t, s_{t+1}^{\star}) - r^D(s_t, a_t, s_{t+1})$

Figure 5: The joint distributions of the model transition MAE, and the dynamics reward $r^D(\cdot)$ for both model transitions and true transitions in the `walker-med-rep` task. The $x$-axes (*model transition MAE*) all denote the MAE between the true transition and the model transition: $\|s_{t+1} - s_{t+1}^{\star}\|_1$.

**Performance of the Dynamics Reward.** To answer **Q1**, we design experiments to evaluate whether the dynamics reward learned by Algorithm 2 can assign transitions proper rewards that are well-aligned with their accuracy. In particular, we choose the `walker-med-rep` task and apply a policy learned by behaviour cloning in the learned dynamics model to collect model rollouts

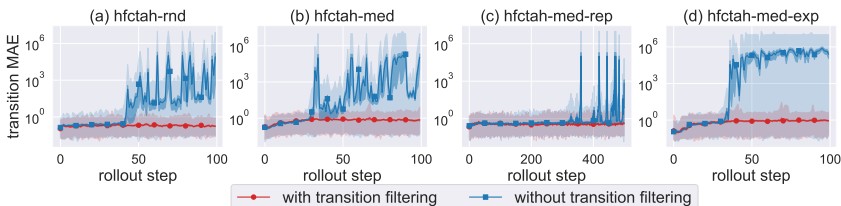

Figure 6: The MAE in terms of rollout steps with or without transition filtering technique.

$\{(s_t, a_t, s_{t+1})\}_t$ with 500 trajectories. Then we record the dynamics reward $r^D(s_t, a_t, s_{t+1})$ and MAE $\|s_{t+1} - s_{t+1}^\star\|_1$ on the collected model rollouts, where $s_{t+1}^\star$ is the ground-truth next state. The joint distribution of $\{(\|s_{t+1} - s_{t+1}^\star\|_1, r^D(s_t, a_t, s_{t+1}))\}_t$ is plotted in Fig. 5a. We observe a strong negative correlation between the dynamics reward and the MAE. The detailed Pearson correlation between the dynamics reward and the model transition MAE is $-0.761$, which also denotes a strong negative correlation between both variables. Besides, we remark that the RMSE on the offline dataset is $0.63$ (refer to Table 12 in Appendix F.5), and thus the transitions with extremely large MAE (e.g., $3.0$) could be regarded as out of distribution data. Even on such OOD transitions, the learned dynamics reward still can assign proper rewards that are consistent with their accuracy, demonstrating its generalization ability. These results clearly demonstrate that the learned dynamics reward is an excellent indicator of the fidelities of transitions.

Moreover, we calculate dynamics rewards on the *true* transitions $r^D(s_t, a_t, s_{t+1}^\star)$. The joint distributions of $\{(\|s_{t+1} - s_{t+1}^\star\|_1, r^D(s_t, a_t, s_{t+1}^\star))\}$ and $\{(\|s_{t+1} - s_{t+1}^\star\|_1, r^D(s_t, a_t, s_{t+1}^\star) - r^D(s_t, a_t, s_{t+1}))\}$ are reported in Fig. 5b and Fig. 5c, respectively. We see that the learned dynamics reward can accurately assign high values to most true transitions. Note that the transitions $(s_t, a_t, s_{t+1})$ and $(s_t, a_t, s_{t+1}^\star)$ only differ in the next state. Nevertheless, the dynamics reward gives totally different values, which suggests that it accurately pays attention to the next state transition. In summary, we empirically verify that the dynamics reward learned by Algorithm 2 is capable of assigning transitions proper rewards that are well-aligned with their accuracy, and thus identifying high-fidelity transitions from all candidates. More visualizations are presented in Appendix F.9.

**Effectiveness of the Transition Filtering Technique.** To answer **Q2**, we conduct an ablation study to verify the effectiveness of utilizing the learned dynamics reward to filter transitions, which is detailed in Eq.(4). Here we choose the `hfctah` tasks where there is no terminal state, which allows the rollout horizon to reach the preset value. We rollout the policy learned by MOREC-MOPO in the learned dynamics model. In particular, we consider two rollout processes: one with the transition filtering technique and the other one without the technique. We respectively take these two rollout processes to collect 100 trajectories and show the MAE on the collected trajectories in Fig. 6. On the one hand, when equipped with the transition filtering technique, the generated model rollouts always keep a relatively small MAE as the rollout step increases. On the other hand, without the transition filtering technique, the MAE on the generated model rollouts blows up as the rollout step increases. This phenomenon clearly demonstrates that utilizing the dynamics reward to filter transitions can effectively reduce compounding errors, and thus provide high-fidelity model rollouts for policy learning. An extended ablation study can be found in Appendix F.7.

## 5 CONCLUSION

In this paper, we propose the MOREC method based on the idea of reward-consistent models. MOREC learns a generalizable dynamics reward which is then used as a transition filter in most offline MBRL methods. We evaluate MOREC through extensive experiments. MOREC outperforms prior methods in 18 out of 21 tasks from the offline RL benchmarks. We empirically validate that the recovered dynamics reward is well-aligned with the accuracy of model transitions even on OOD regions, demonstrating its generalization ability. Consequently, utilizing such a generalizable dynamics reward to filter transitions can effectively reduce compounding errors. However, we also noted the discriminator ensemble could consume a significant memory, especially in the tasks with large state and action dimensions, potentially limiting the scalability of MOREC. A future direction is adopting the last-iterate convergence learning methods (Lei et al., 2021; Golowich et al., 2020), which keep only the last discriminator while maintaining similar convergence guarantees.

ACKNOWLEDGEMENTS

The authors would like to thank Dr. Zuolin Tu and anonymous reviewers for their constructive advice to improve this paper. This work was supported by the National Science Foundation of China (61921006).

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

CONTENTS

# A  EXTENDED RELATED WORK

**Offline RL.** Offline RL studies the methodologies that enable the agent to directly learn an effective policy from an offline experience dataset without any interaction with the environment dynamics. The major challenge of offline RL arises from the discrepancy between the offline experience dataset and the behavior policy's visitation, resulting in extrapolation errors (Fujimoto et al., 2019; Kumar et al., 2019). Model-free offline RL algorithms (Fujimoto et al., 2019; Fujimoto & Gu, 2021; Kostrikov et al., 2021; Kumar et al., 2019; 2020; Bai et al., 2022) incorporate conservatism into policy or Q-function of online RL algorithms to tackle with extrapolation error.

The challenge arises from the discrepancy between the offline experience dataset and the behavior policy's visitation. This mismatch can lead to a poor estimation of the Q-function of the behavior policy, plagued by extrapolation errors (Fujimoto et al., 2019; Kumar et al., 2019). Consequently, online off-policy RL algorithms fail to be applied in offline RL directly.

To tackle extrapolation error, conservatism is introduced to offline RL algorithms as a common paradigm. Model-free offline RL algorithms incorporate conservatism or regularization into online RL algorithms by preventing the behavior policy from acting in out-of-support regions (Fujimoto et al., 2019; Fujimoto & Gu, 2021; Kostrikov et al., 2021) or by learning a conservative Q-function for out-of-distribution (OOD) visitation of the behavior policy (Kumar et al., 2019; 2020; Bai et al., 2022), without learning dynamics models.

**Offline MBRL.** Offline MBRL algorithms (Yu et al., 2020; Kidambi et al., 2020; Chen et al., 2021; Yu et al., 2021) leverage a dynamics model derived from offline data to enhance the efficiency of offline RL. Benefiting from the additional synthetic data generated by the learned dynamics model, model-based algorithms are more likely to have the potential to generalize in the states out of distribution and perform better in new tasks. However, the learned dynamics model also suffers from errors inevitably for the limited experience dataset (Xu et al., 2022a; Janner et al., 2019). Thus, some works (Yu et al., 2020; Sun et al., 2023; Kidambi et al., 2020; Yu et al., 2021) also incorporate conservatism into offline MBRL. For example, some methods (Yu et al., 2020; Bai et al., 2022; Sun et al., 2023) apply uncertainty estimation and trust states with low uncertainty, while some methods (Kidambi et al., 2020; Yu et al., 2021) try to limit the behavior policy to acting surrounding the experience dataset. However, despite these efforts, the performance of these techniques is largely dependent on the accuracy of the dynamics model.

**Inverse Reinforcement Learning (IRL)** IRL (Ng & Russell, 2000; Ni et al., 2020; Ghasemipour et al., 2019; Swamy et al., 2021) is a process that tackles MDPs devoid of reward functions. The aim of IRL is to deduce these reward functions from a series of expert demonstrations. Apprenticeship learning (Abbeel & Ng, 2004) trains reward by maximizing an evaluation margin between the expert and the policy. In MaxEnt IRL algorithm, the reward is modeled as a maximum likelihood problem by introducing a maximum entropy objective (Ziebart et al., 2008). GCL (Finn et al., 2016b) extends MaxEnt IRL to high-dimensional state-action space. Recently, as a special variant of IRL, adversarial imitation learning (AIL) (Ho & Ermon, 2016; Finn et al., 2016a; Fu et al., 2017; Kostrikov et al., 2020; Sun et al., 2021; Kostrikov et al., 2019) learns policy and reward in a full adversarial manner.

**Generalizable reward learning** (Fu et al., 2017), which aims at learning reward functions robust to the dynamics changes, have attracted lots of interests recently. Generalizable reward learning enable the learned reward to be reusable in a dynamics-mismatch environment, thus largely extending the scope of IRL applications. Prior works have shown such a generalizable reward can be recovered under some assumptions (Geng et al., 2020; Amin & Singh, 2016; Jacq et al., 2019), e.g. assuming the true reward is only related to the state (Fu et al., 2017). Besides, Peng et al. (2019) also show that, by incorporating regularization like mutual information constraints into reward learning, we can largely improve the generalizability of the reward functions. In this paper, we also introduce regularization to enhance the generalizability of the dynamics reward.

Despite previous efforts, the performance of offline MBRL algorithms is intrinsically limited by the prediction accuracy of the learned dynamics model itself. In this work, with the aim of obtaining more accurate predictions of dynamics, we employ a better generalizable reward of dynamics model, which is learned by IRL, to correct the transitions during the policy learning phase in an offline MBRL algorithm.

# B   ADDITIONAL RESULTS FOR DYNAMICS REWARD LEARNING

## B.1   DYNAMICS REWARD LEARNING ALGORITHM

Here we present the completed algorithm for dynamics reward learning. Let $\mathcal{W} = \{D \in \mathbb{R}^{|\mathcal{S}|^2|\mathcal{A}|} :$ $0 < D(s,a,s') < 1, \forall(s,a,s') \in \mathcal{S} \times \mathcal{A} \times \mathcal{S}\}$ and $\mathcal{P} = \{P \in \mathbb{R}^{|\mathcal{S}|^2|\mathcal{A}|} : P(s'|s,a) \geq 0, \forall(s,a,s') ; \sum_{s' \in \mathcal{S}} P(s'|s,a) = 1, \forall(s,a) \in \mathcal{S} \times \mathcal{A}\}$ denote the sets of all feasible discriminators and dynamics models, respectively. Besides, for a set $\mathcal{X} \subseteq \mathbb{R}^d$, we use $\Pi_{\mathcal{X}}$ to denote the $\ell_2$-norm-based projection operator onto the set $\mathcal{X}$, i.e., $\Pi_{\mathcal{X}}(y) = \text{argmin}_{x \in \mathcal{X}} \|x - y\|_2^2$.

---

**Algorithm 2:** Dynamics Reward Learning

---

**Input:** Offline data $\mathcal{D}$; number of iterations $T$; step sizes $\{\eta_t^D, \eta_t^P\}_{t=1}^T$

1  Initialize the discriminator $D_1$ and dynamics model $P_1$;

2  **for** $t = 1, \ldots, T$ **do**

3  $\quad D_{t+1} \leftarrow \Pi_{\mathcal{W}}\Big(D_t + \eta_t^D\big(\mathbb{E}_{(s,a,s') \sim \mathcal{D}}\left[\nabla_D \log\left(D_t(s,a,s')\right)\right] +$

$\qquad \mathbb{E}_{(s,a) \sim \mathcal{D}, s' \sim P_t(\cdot|s,a)}\left[\nabla_D \log\left(1 - D_t(s,a,s')\right)\right]\big)\Big);$

4  $\quad P_{t+1} \leftarrow \Pi_{\mathcal{P}}\big(P_t - \eta_t^P \mathbb{E}_{(s,a) \sim \mathcal{D}, s' \sim P(\cdot|s,a)}\left[\nabla_P \log\left(P_t(s'|s,a)\right) \log\left(1 - D_t(s,a,s')\right)\right]\big);$

**Output:** The reward model: $r(s,a,s') = -\log(1 - \sum_{t=1}^T \log(1 - D_t(s,a,s'))/T)$

---

## B.2   PROOF OF PROPOSITION 1

To prove Proposition 1, we need the following auxiliary Lemma.

**Lemma 1.** *Consider the objective* $\ell(D,P) = \mathbb{E}_{(s,a,s') \sim \mathcal{D}}\left[\log(D(s,a,s'))\right] + \mathbb{E}_{(s,a) \sim \mathcal{D}, s' \sim P(\cdot|s,a)}\left[\log(1 - D(s,a,s'))\right]$. $\ell(D,P)$ *is concave in* $D \in \mathcal{W}$ *and convex in* $P \in \mathcal{P}$, *where* $\mathcal{W} = \{D \in \mathbb{R}^{|\mathcal{S}|^2|\mathcal{A}|} : 0 < D(s,a,s') < 1, \forall(s,a,s') \in \mathcal{S} \times \mathcal{A} \times \mathcal{S}\}$ *and* $\mathcal{P} = \{P \in \mathbb{R}^{|\mathcal{S}|^2|\mathcal{A}|} : P(s'|s,a) \geq 0, \forall(s,a,s') ; \sum_{s' \in \mathcal{S}} P(s'|s,a) = 1, \forall(s,a) \in \mathcal{S} \times \mathcal{A}\}$. *Besides,* $\forall D, D' \in \mathcal{W}, \|D - D'\|_2 \leq 2\sqrt{|\mathcal{S}|^2|\mathcal{A}|}, \forall P, P' \in \mathcal{P}, \|P - P'\|_2 \leq 2\sqrt{|\mathcal{S}||\mathcal{A}|}$.

*Proof of Lemma 1.* First, we apply the second-order condition (Boyd & Vandenberghe, 2004) to verify the concavity of $\ell(D,P)$ with respect to $D$. In particular, we calculate the gradient $\nabla_D \ell(D,P) \in \mathbb{R}^{|\mathcal{S}|^2|\mathcal{A}|}$ with the element

$$\nabla_{D(s,a,s')}\ell(D,P) = \widehat{d}(s,a,s')\frac{1}{D(s,a,s')} - \widehat{d}(s,a)P(s'|s,a)\frac{1}{1 - D(s,a,s')}.$$

With a slight abuse of notations, we use $\widehat{d}(s,a,s')$ and $\widehat{d}(s,a)$ to denote the empirical distributions from the offline dataset $\mathcal{D}$. Concretely,

$$\widehat{d}(s,a,s') = \frac{\sum_{(s_t,a_t,s_{t+1}) \in \mathcal{D}} \mathbb{I}\{s_t = s, a_t = a, s_{t+1} = s'\}}{|\mathcal{D}|}, \widehat{d}(s,a) = \frac{\sum_{(s_t,a_t) \in \mathcal{D}} \mathbb{I}\{s_t = s, a_t = a\}}{|\mathcal{D}|}.$$

We further calculate the Hessian matrix $\nabla_D^2 \ell(D,P) \in \mathbb{R}^{|\mathcal{S}|^2|\mathcal{A}| \times |\mathcal{S}|^2|\mathcal{A}|}$. We note that $\nabla_D^2 \ell(D,P)$ is a diagonal matrix with the principal diagonal elements of

$$\nabla_{D(s,a,s')}^2 \ell(D,P) = -\widehat{d}(s,a,s')\frac{1}{D(s,a,s')^2} - \widehat{d}(s,a)P(s'|s,a)\frac{1}{(1 - D(s,a,s'))^2} < 0.$$

Therefore, the Hessian matrix $\nabla_D^2 \ell(D,P) \in \mathbb{R}^{|\mathcal{S}|^2|\mathcal{A}| \times |\mathcal{S}|^2|\mathcal{A}|}$ is a negative definite matrix. Based on the second-order condition (Boyd & Vandenberghe, 2004), we have that $\ell(D,P)$ is concave in $D$. Besides, for any fixed $D \in \mathcal{W}$, $\ell(D,P)$ is a linear function with respect to $P$. Therefore, $\ell(D,P)$ is convex in $P$.

Furthermore, $\forall D, D' \in \mathcal{W}$, we have that $\|D - D'\|_2 \leq 2\|D\|_2 \leq 2\sqrt{|\mathcal{S}|^2|\mathcal{A}|}$. Similarly, we have that $\forall P, P' \in \mathcal{W}$, $\|P - P'\|_2 \leq 2\|P\|_2 = 2\sqrt{\sum_{(s,a)\in\mathcal{S}\times\mathcal{A}}\sum_{s'} P(s'|s,a)^2} \leq 2\sqrt{\sum_{(s,a)\in\mathcal{S}\times\mathcal{A}}(\sum_{s'} P(s'|s,a))^2} = 2\sqrt{|\mathcal{S}||\mathcal{A}|}$. We finish the proof. $\qquad\square$

Now we proceed to prove Proposition 1. We have that

$$f\left(\frac{1}{T}\sum_{t=1}^{T} D_t\right) = \min_{P\in\mathcal{P}} \ell\left(\frac{1}{T}\sum_{t=1}^{T} D_t, P\right) \geq \min_{P\in\mathcal{P}} \frac{1}{T}\sum_{t=1}^{T} \ell(D_t, P).$$

The last inequality follows that $\ell(D, P)$ is concave in $D$ from Lemma 1. Notice that the variable $P$ takes the projected gradient descent updates with respect to the sequence of functions $\{\ell(D_t, P)\}_{t=1}^{T}$. From Lemma 1, we have that $\forall t \in [T]$, $\ell(D_t, P)$ is convex in $P$ and $\forall P, P' \in \mathcal{P}, \|P - P'\|_2 \leq 2\sqrt{|\mathcal{S}||\mathcal{A}|}$. Furthermore, we suppose that the gradient norm is bounded, i.e., $\|\nabla_P \ell(D_t, P_t)\|_2 \leq G^P$. We can apply Theorem 3.1 in (Hazan, 2016) to obtain

$$\min_{P\in\mathcal{P}} \frac{1}{T}\sum_{t=1}^{T} \ell(D_t, P) \geq \frac{1}{T}\sum_{t=1}^{T} \ell(D_t, P_t) - 3G^P\sqrt{\frac{|\mathcal{S}||\mathcal{A}|}{T}}.$$

Then we arrive at

$$f\left(\frac{1}{T}\sum_{t=1}^{T} D_t\right) \geq \frac{1}{T}\sum_{t=1}^{T} \ell(D_t, P_t) - 3G^P\sqrt{\frac{|\mathcal{S}||\mathcal{A}|}{T}}.$$

Similarly, the variable $D$ takes the projected gradient ascent updates with respect to the sequence of functions $\{\ell(D, P_t)\}_{t=1}^{T}$. From Lemma 1, we obtain that $\forall t \in [T]$, $\ell(D, P_t)$ is concave in $D$ and $\forall D, D' \in \mathcal{D}, \|D - D'\|_2 \leq 2\sqrt{|\mathcal{S}|^2|\mathcal{A}|}$. Besides, we suppose that the gradient norm is bounded, i.e., $\|\nabla_D \ell(D_t, P_t)\|_2 \leq G^D$. Applying Theorem 3.1 in (Hazan, 2016) yields that

$$\frac{1}{T}\sum_{t=1}^{T} \ell(D_t, P_t) \geq \max_{D\in\mathcal{W}} \frac{1}{T}\sum_{t=1}^{T} \ell(D, P_t) - 3G^D\sqrt{\frac{|\mathcal{S}|^2|\mathcal{A}|}{T}}.$$

We combine the above two inequalities and get that

$$f\left(\frac{1}{T}\sum_{t=1}^{T} D_t\right) \geq \max_{D\in\mathcal{W}} \frac{1}{T}\sum_{t=1}^{T} \ell(D, P_t) - 3G^P\sqrt{\frac{|\mathcal{S}||\mathcal{A}|}{T}} - 3G^D\sqrt{\frac{|\mathcal{S}|^2|\mathcal{A}|}{T}}$$

$$\geq \max_{D\in\mathcal{W}} \min_{P\in\mathcal{P}} \ell(D, P) - 3G^P\sqrt{\frac{|\mathcal{S}||\mathcal{A}|}{T}} - 3G^D\sqrt{\frac{|\mathcal{S}|^2|\mathcal{A}|}{T}}$$

$$= \max_{D\in\mathcal{W}} f(D) - 3G^P\sqrt{\frac{|\mathcal{S}||\mathcal{A}|}{T}} - 3G^D\sqrt{\frac{|\mathcal{S}|^2|\mathcal{A}|}{T}}.$$

We complete the proof.

### B.3 EXTENSION OF PROPOSITION 1 TO LINEAR FUNCTION APPROXIMATION

Proposition 1 considers the tabular setting where the discriminator and dynamics model are directly represented by tables. In this part, we extend Proposition 1 to the linear function approximation setting. In this setting, based on the linear mixture MDP formulation (Ayoub et al., 2020; Cai et al., 2020), we consider that the discriminator and dynamics model are represented by linear functions. Formally, we have that

$$\forall (s, a, s') \in \mathcal{S} \times \mathcal{A} \times \mathcal{S}, \; D_\omega(s, a, s') = \langle \phi(s, a, s'), \omega \rangle, \; P_\nu(s'|s, a) = \langle \phi(s, a, s'), \nu \rangle.$$

Here $\phi : \mathcal{S} \times \mathcal{A} \times \mathcal{S} \to \mathbb{R}^d$ is the known feature function and $\omega, \nu \in \mathbb{R}^d$ are parameters to be learned. To ensure that the parameterized discriminators and dynamics models are feasible, we consider the following parameter spaces.

$$\mathcal{W} = \left\{ \omega \in \mathbb{R}^d : \|\omega\|_2 \leq \sqrt{d}, \; \forall (s, a, s') \in \mathcal{S} \times \mathcal{A} \times \mathcal{S}, \; 0 < D_\omega(s, a, s') < 1 \right\},$$

$$\mathcal{V} = \left\{ \nu \in \mathbb{R}^d : \|\nu\|_2 \leq \sqrt{d}, \ \forall (s, a, s') \in \mathcal{S} \times \mathcal{A} \times \mathcal{S}, \ P_\nu(s'|s, a) \geq 0, \sum_{s' \in \mathcal{S}} P_\nu(s'|s, a) = 1 \right\}.$$

Following (Ayoub et al., 2020; Cai et al., 2020), we consider that the $\ell_2$-norm of parameters is bounded by $\sqrt{d}$. In this linear function approximation setting, we consider the optimization problem in the parameter space.

$$\max_{\omega \in \mathcal{W}} \min_{\nu \in \mathcal{V}} \ell(\omega, \nu) := \mathbb{E}_{(s,a,s') \sim \mathcal{D}} \left[ \log(D_\omega(s, a, s')) \right] + \mathbb{E}_{(s,a) \sim \mathcal{D}, s' \sim P_\nu(\cdot|s,a)} \left[ \log(1 - D_\omega(s, a, s')) \right].$$

Accordingly, we can define the function $f(\omega) := \min_{\nu \in \mathcal{V}} \ell(\omega, \nu)$.[1]

For the algorithm, we still consider the gradient-descent-ascent method but the optimization is performed in the parameter space instead of the function space in the tabular setting.

---

**Algorithm 3:** Dynamics Reward Learning with Linear Function Approximation

---

**Input:** Offline data $\mathcal{D}$; number of iterations $T$; step sizes $\{\eta_t^\omega, \eta_t^\nu\}_{t=1}^T$

1 Initialize the discriminator parameter $\omega_1$ and dynamics model parameter $\nu_1$;

2 **for** $t = 1, \ldots, T$ **do**

3 $\quad \omega_{t+1} \leftarrow \Pi_{\mathcal{W}} \Big( \omega_t + \eta_t^\omega \big( \mathbb{E}_{(s,a,s') \sim \mathcal{D}} [\nabla_\omega \log(D_{\omega_t}(s, a, s'))] +$

$\quad \mathbb{E}_{(s,a) \sim \mathcal{D}, s' \sim P_{\nu_t}(\cdot|s,a)} [\nabla_\omega \log(1 - D_{\omega_t}(s, a, s'))] \big) \Big)$;

4 $\quad \nu_{t+1} \leftarrow \Pi_{\mathcal{V}} \big( \nu_t - \eta_t^\nu \mathbb{E}_{(s,a) \sim \mathcal{D}, s' \sim P_{\nu_t}(\cdot|s,a)} [\nabla_\nu \log(P_{\nu_t}(s'|s,a)) \log(1 - D_{\omega_t}(s, a, s'))] \big)$;

**Output:** The reward model: $r(s, a, s') = -\log(1 - \sum_{t=1}^T \log(1 - D_{\omega_t}(s, a, s'))/T)$

---

**Proposition 2.** *Consider Algorithm 3. Let $\Phi \in \mathbb{R}^{|\mathcal{D}| \times d}$ be the feature matrix that aggregates the feature vectors of transitions in $\mathcal{D}$. Assume that $\mathbf{rank}(\Phi) = d$. Further assume that the gradient norm is bounded, i.e., $\forall t \in [T], \|\nabla_\nu \ell(\omega_t, \nu_t)\|_2 \leq G^\nu, \|\nabla_\omega \ell(\omega_t, \nu_t)\|_2 \leq G^\omega$. If we take the step size of $\{\eta_t^\nu = \sqrt{d}/(G^\nu t), \eta_t^\omega = \sqrt{d}/(G^\omega t)\}_{t=1}^T$, then we have*

$$f\left( \frac{1}{T} \sum_{t=1}^T \omega_t \right) \geq \max_{\omega \in \mathcal{W}} f(\omega) - 3G^\omega \sqrt{\frac{d}{T}} - 3G^\nu \sqrt{\frac{d}{T}}.$$

*Proof.* The proof idea is similar to that in Appendix B.2. First, we prove that the objective $\ell(\omega, \nu)$ is concave-convex with respect to $(\omega, \nu)$. In particular, we can calculate the Hessian matrix of $\ell(\omega, \nu)$ regarding $\omega$.

$$\nabla_\omega^2 \ell(\omega, \nu) = -\mathbb{E}_{(s,a,s') \sim \mathcal{D}} \left[ \frac{1}{D_\omega^2(s, a, s')} \phi(s, a, s') \phi^\top(s, a, s') \right]$$

$$- \mathbb{E}_{(s,a) \sim \mathcal{D}, s' \sim P_\nu(\cdot|s,a)} \left[ \frac{1}{(1 - D_\omega^2(s, a, s'))^2} \phi(s, a, s') \phi^\top(s, a, s') \right].$$

For any non-zero vector $x \in \mathbb{R}^d$, we have that

$$x^\top \nabla_\omega^2 \ell(\omega, \nu) x = -\mathbb{E}_{(s,a,s') \sim \mathcal{D}} \left[ \frac{1}{D_\omega^2(s, a, s')} (\phi^\top(s, a, s') x)^2 \right]$$

$$- \mathbb{E}_{(s,a) \sim \mathcal{D}, s' \sim P_\nu(\cdot|s,a)} \left[ \frac{1}{(1 - D_\omega^2(s, a, s'))^2} (\phi^\top(s, a, s') x)^2 \right].$$

Notice that we consider the under-parameterization case where $\mathbf{rank}(\Phi) = d$. With the rank theorem, we have that $\dim(\mathcal{N}(\Phi)) = 0$, where $\mathcal{N}(\Phi)$ denotes the nullspace of $\Phi$. This implies that there exists $(s, a, s)' \in \mathcal{D}$ such that $\phi^\top(s, a, s') x \neq 0$. Therefore, we obtain that $x^\top \nabla_\omega^2 \ell(\omega, \nu) x < 0$, indicating that $\nabla_\omega^2 \ell(\omega, \nu)$ is a negative definite matrix. With the second-order condition of concavity, we obtain

---

[1] Here we still use the notations $\ell$ and $f$ to denote the functions in the parameter space when the context is clear.

that $\ell(\omega, \nu)$ is concave with respect to $\omega$. Moreover, it is straightforward to observe that for any fixed $\omega \in \mathcal{W}$, $\ell(\omega, \nu)$ is a linear function with respect to $\nu$. Thus, $\ell(\omega, \nu)$ is convex regarding $\nu$.

Furthermore, it is easy to show that the radiuses of $\mathcal{W}$ and $\Omega$ are bounded. That is, $\forall \omega, \omega' \in \mathcal{W}$ and $\forall \nu, \nu' \in \mathcal{V}$, $\|\omega - \omega'\|_2 \leq 2\sqrt{d}$, $\|\nu - \nu'\|_2 \leq 2\sqrt{d}$. In summary, we have proved that $\ell(\omega, \nu)$ is concave-convex regarding $(\omega, \nu)$, and the radiuses of $\mathcal{W}$ and $\mathcal{V}$ are bounded by $2\sqrt{d}$.

With these elementary results, we proceed to prove the final convergence result. In particular, we have that

$$
\begin{aligned}
f\left(\frac{1}{T}\sum_{t=1}^{T}\omega_t\right) &\overset{(a)}{\geq} \min_{\nu \in \mathcal{V}} \frac{1}{T}\sum_{t=1}^{T}\ell(\omega_t, \nu) \\
&\overset{(b)}{\geq} \frac{1}{T}\sum_{t=1}^{T}\ell(\omega_t, \nu_t) - 3G^{\nu}\sqrt{\frac{d}{T}} \\
&\overset{(c)}{\geq} \max_{\omega \in \mathcal{W}} \frac{1}{T}\sum_{t=1}^{T}\ell(\omega, \nu_t) - 3G^{\omega}\sqrt{\frac{d}{T}} - 3G^{\nu}\sqrt{\frac{d}{T}}
\end{aligned}
$$

Here inequality $(a)$ follows the concavity of $\ell(\nu, \omega)$ in $\nu$. Moreover, notice that the parameter $\nu$ is updated by performing projected gradient descent regarding the sequence of functions $\{\ell(\omega_t, \nu)\}_{t=1}^{T}$. Recall that $\ell(\omega, \nu)$ is convex in $\nu$ and the radius of $\mathcal{V}$ is bounded by $2\sqrt{d}$. Then we can apply (Hazan, 2016, Theorem 3.1), resulting in inequality $(b)$. Similarly, the parameter $\nu$ takes the projected gradient ascent updates with respect to the sequence of functions $\{\ell(\omega, \nu_t)\}_{t=1}^{T}$. Note that $\ell(\omega, \nu)$ is concave in $\omega$ and the radius of $\mathcal{W}$ is bounded by $2\sqrt{d}$. We can leverage (Hazan, 2016, Theorem 3.1) to obtain inequality $(c)$.

Then we can get that

$$
\begin{aligned}
f\left(\frac{1}{T}\sum_{t=1}^{T}\omega_t\right) &\geq \max_{\omega \in \mathcal{W}} \frac{1}{T}\sum_{t=1}^{T}\ell(\omega, \nu_t) - 3G^{\omega}\sqrt{\frac{d}{T}} - 3G^{\nu}\sqrt{\frac{d}{T}} \\
&\geq \max_{\omega \in \mathcal{W}} \min_{\nu \in \mathcal{V}} \ell(\omega, \nu) - 3G^{\omega}\sqrt{\frac{d}{T}} - 3G^{\nu}\sqrt{\frac{d}{T}} \\
&= \max_{\omega \in \mathcal{W}} f(\omega) - 3G^{\omega}\sqrt{\frac{d}{T}} - 3G^{\nu}\sqrt{\frac{d}{T}},
\end{aligned}
$$

which finishes the proof. $\qquad\square$

## C  CONNECTIONS BETWEEN MOREC AND ADVERSARIAL MODEL LEARNING

Adversarial model learning employs the generative adversarial imitation learning framework to learn the dynamics model (Shi et al., 2019; Xu et al., 2022a). It also uses the objective as Eq.Eq.(2). The equivalence between adversarial imitation learning and inverse reinforcement learning (Finn et al., 2016a) highlights the shared principle with MOREC, which explicitly learns the reward model. Both methods benefit from small compounding error (Xu et al., 2022a) and exhibit causality-consistent transitions (Chen et al., 2023).

However, there are noticeable differences. MOREC employs an ensemble of discriminators as the reward model, which is more stable and generalizable compared to using a single discriminator as in previous adversarial model learning methods (Luo et al., 2022). Additionally, the filtering mechanism allows supervised and adversarial model learning to complement each other. This is particularly advantageous for large training data, as supervised model learning helps MOREC leverage all the available data. In contrast, previous adversarial model learning methods solely rely on generative adversarial learning, which may fail to capture small modes in the data.

# D  IMPLEMENTATION DETAILS

## D.1  DETAILED IMPLEMENTATION OF REWARD LEARNING

We adopt the dynamics reward learning approach grounded on the GAIL framework (Ho & Ermon, 2016). As delineated in Section 3.2 and Algorithm 2, this process involves two main iterative stages: the discriminator updating stage and the dynamics model updating stage.

During the discriminator updating stage (refer to line 3 in Algorithm 2), the parameters of the discriminator are updated using a single gradient step. This step utilizes both the offline dataset and data procured from the current dynamics model. Conversely, in the dynamics model updating phase, data is sampled from the current model to compute rewards based on $\log(1 - D_t(s, a, s'))$. The dynamics model parameters are subsequently refined through an advanced policy gradient method, namely PPO (Schulman et al., 2017). To ensure stable learning dynamics between the discriminator and the dynamics model, the updating of the dynamics model and the discriminator model is repeated 5 times ($\texttt{d\_step} = \texttt{g\_step} = 5$).

Additionally, beyond the standard two-term discriminator loss as depicted in Eq.(2), we integrate a gradient penalty to regularize the discriminator, thus curtailing overfitting on offline data. The final discriminator's optimization objective is represented as:

$$\tilde{\ell}(D, P) = \ell(D, P) + \eta \|\nabla_D \ell(D, P)\|_2^2, \tag{5}$$

where $\eta$ serves as a regularization coefficient.

To further optimize computational efficiency, we cap the stored discriminators to a maximum count of $L$ and archive the discriminator at every $H$ iterations. For tasks in D4RL and NeoRL, the parameters are set at $L = 40$ and $H = 10$. However, for the refrigerator task, we use $L = 200$ and $H = 1$. Other hyper-parameters used for dynamics reward learning are listed in Table 4.

Table 4: The hyper-parameters of the dynamics reward learning.

| Attribute | Value |
|---|---|
| Number of training iteration | 5000 |
| Batch size per PPO epoch | 20000 |
| Discriminator learning rate | 1e-3 |
| Adversarial dynamics model learning rate | 3e-4 |
| Value network learning rate | 1e-3 |
| Optimizer | Adam (Kingma & Ba, 2015) |
| Hidden layers of the adversarial dynamics model | [256, 256] |
| Hidden layers of the value function | [256, 256] |
| Hidden layers of the discriminator | [128, 256, 256, 128] |
| $\eta$ | 0.75 for $\texttt{HalfCheetah}$ in NeoRL and 0.5 otherwise |

## D.2  DETAILED IMPLEMENTATION OF THE UNCERTAINTY ESTIMATION IN MOREC-MOPO

The proposed MOREC-MOPO technique commences by training an ensemble of dynamics models, denoted as $\mathcal{P} = \{p_{\theta_i}(s'|s, a) \mid i \in [M]\}$. Each individual model within this ensemble, specifically $p_{\theta_i}(s'|s, a)$, is characterized as a Gaussian distribution, which is parameterized by a neural network $g(s, a; \theta_i)$. The output of this neural network is splited into two components: the mean $g^\mu(s, a; \theta_i)$ and the standard deviation $g^\sigma(s, a; \theta_i)$. Thus, the dynamics model can be explicitly expressed as:

$$p_{\theta_i}(s'|s, a) = \mathcal{N}(\cdot|g^\mu(s, a; \theta_i), g^\sigma(s, a; \theta_i)^2).$$

To estimate the uncertainty, we employ the concept of max aleatoric (Yu et al., 2020). This term depends on the maximum aleatoric error. In a mathematical context, the aleatoric uncertainty for each model is defined as the L2-norm of its standard deviation, represented as $\|g^\sigma(s, a; \theta_i)\|_2$ for the $i$-th model. Therefore, the maximal aleatoric error is formalized in Eq.(6):

$$\mathcal{U}(\mathcal{P}, s_t, a_t, s_{t+1}) = \max_i \|g^\sigma(s, a; \theta_i)\|_2. \tag{6}$$

### D.3 DETAILED IMPLEMENTATION OF MOREC

We developed MOREC using the `OfflineRL-Kit` codebase (Sun, 2023). From this base, we have made the following primary modifications:

1. Initialize the program by loading the pre-trained dynamics reward function.
2. Refine the `step` method within the dynamics model class:
   (a) Execute sampling for each model within the ensemble $N$ times.
   (b) Compute the dynamics rewards for the obtained samples.
   (c) Formulate a softmax distribution and sample an index from this distribution.
   (d) Return the next state corresponding to the sampled index.

## E EXPERIMENT DETAILS

### E.1 DETAILED EXPERIMENT SETTING

**The refrigerator temperature-control task.** In the refrigerator temperature-control task, an agent controls the compressor's power in a refrigerator with the primary aim of maintaining a set target temperature. The agent's observations is the present temperature, with the action corresponding to the normalized power of the compressor.

The system employs a transition function $f(\text{temp}, a, z) : \mathbb{R} \times \mathbb{R} \times \{0, 1\} \to \mathbb{R}$ that operates in two distinct modes. The modes indicate the current state of the refrigerator door: open ($z = 1$) or closed ($z = 0$). These modes are characterized by different rates of temperature change, with the open-door state causing a more rapid convergence towards room temperature (15 °C). The transition function, as detailed in Eq.(7), formalizes this behavior:

$$f(\text{temp}_t, a_t, z_t) = \text{temp}_t - a + (0.02 + 0.06z)(15 - \text{temp}_t). \tag{7}$$

Fig. 2 (a) graphically illustrates the dynamics of this system when the refrigerator door is closed ($z = 0$), mapping temperature, action, and the next temperature on the x, y, and color axes respectively.

We generated offline data using a proportional controller as the behavior policy, simulating 2000 interactions in an environment where the refrigerator door periodically opens and closes. This policy, formalized in Eq.(8), is also depicted in Fig. 2 (a):

$$\pi^{\text{sample}}(a_t \mid \text{temp}_t) = \mathcal{N}(a_t \mid 0.2\text{temp}_t + 0.75, 0.1). \tag{8}$$

In the policy learning stage (Fig. 8, 11, and Table 1, 9), we designed a temperature-control task. The corresponding reward is set to

$$r(\text{temperature}_t) = -|\text{temperature}_t + 1|, \tag{9}$$

where temperature$_t$ is the temperature of the $t$-th step.

**The NeoRL tasks.** We installed NeoRL from the official repository[2] and used its 1000-trajectory offline dataset to accomplish all NeoRL experiments.

### E.2 HYPER-PARAMETERS

The hyper-parameters for MOREC-MOPO and MOREC-MOBILE derive from the default parameters specified in MOPO[3] and MOBILE in `OfflineRL-Kit`[4]. Modifying these defaults, we extend the rollout horizon from either 1 or 5 to 100, while diminishing the number of rollouts per epoch to alleviate computational strain. This adaptation leads to the consolidated hyper-parameters for both MOREC-MOPO and MOREC-MOBILE, as detailed in Table 5.

Observing that the model transition MAE consistently exceeds 1.0 when $r_{\min}^D \leq 0.6$ as indicated in Fig. 5a, we establish $r_{\min}^D$ at 0.6. Notably, this 1.0 surpasses all validation root mean square errors

---

[2]https://github.com/polixir/NeoRL
[3]https://github.com/yihaosun1124/OfflineRL-Kit
[4]https://github.com/yihaosun1124/mobile

presented in Table 12. Additionally, a value of $0.6$ demonstrates efficacy across a majority of tasks when applied to MOREC-MOPO and MOREC-MOBILE.

Our investigation primarily revolved around tuning two critical hyper-parameters: the temperature coefficient $\kappa$ and the penalty coefficient $\beta$.

For $\kappa$, the search space is limited to $\{0, 0.1\}$, where $\kappa = 0$ indicates the selection of the subsequent state relying on the maximum dynamics reward.

Regarding $\beta$, we began by identifying the optimal value for MOREC-MOPO within the D4RL tasks. We determined the search space for $\beta$ by adjusting the default penalty coefficient using scaling factors $\{0.1, 0.5, 1.0, 2.0, 3.0, 4.0\}$. In the context of NeoRL, given the absence of a predefined $\beta$ in the `OfflineRL-Kit` repository, we need to search the hyper-parameters in the NeoRL tasks from scratch. Setting the base value of $\beta$ at $2.0$, we then make a search around the base value. It was observed that both `Hopper-v3-H` and `Hopper-v3-M` demand a larger $\beta$, whereas `Walker2d-v3-H` requires a markedly smaller value. The finalized hyper-parameter configurations for MOREC-MOPO can be found in Table 6.

Transitioning to MOREC-MOBILE, we employed a searching strategy akin to that used in MOREC-MOPO. Using the $\beta$ values reported by Sun et al. (2023) as a foundation, we conducted multiple searches around this baseline by adjusting $\beta$ using coefficients such as $[0.3, 0.4, 0.5, 0.75, 1.0, 1.5]$. It was discerned that MOREC-MOBILE exhibits slightly higher sensitivity to variations in $\beta$ compared to MOREC-MOPO. This prompted an additional round of hyper-parameter refinement for some tasks, based on initial search outcomes. The definitive hyper-parameter settings for MOREC-MOBILE are detailed in Table 7.

Table 5: The common hyper-parameters in *MOREC-MOPO* and *MOREC-MOBILE*

| Attribute | Value |
|---|---|
| Actor learning rate | 1e-4 |
| Critic learning rate | 3e-4 |
| Dynamics learning rate | 1e-3 |
| Model ensemble size | 7 |
| Number of the selected models (M in Eq.(4)) | 5 |
| Number of resample times (N in Eq.(4)) | 2 |
| The number of critics | 2 |
| Hidden layers of the actor network | [256, 256] |
| Hidden layers of the critic network | [256, 256] |
| Hidden layers of the dynamics model | [200, 200, 200, 200] |
| Discount factor $\gamma$ | 0.99 |
| Target network smoothing coefficient $\tau$ | 5e-3 |
| Max Rollout horizon $H_{\text{rollout}}$ | 100 |
| Optimizer of the actor and critics | Adam |
| Rollout number per epoch | 2000 |
| Batch size of optimization | 256 |
| Batch number of infering reward | 4 |
| Threshold of dynamics reward for rollout termination ($r^D_{\min}$) | 0.6 |
| Total steps of optimization | 3e6 |
| Actor optimizer learning schedule | Cosine learning schedule |

### E.3 BASELINES

Here we introduce the baselines used in our experiments, including model-free offline RL and model-based offline RL.

**Model-free offline RL.**

* CQL (Kumar et al., 2020) adds penalization to Q-values for the samples out of distribution;
* TD3+BC (Fujimoto & Gu, 2021) incorperates a BC regularization term into the policy optimization objective;
* EDAC (An et al., 2021) proposed to penalize based on the uncertainty degree of the Q-value.

Table 6: Particular hyper-parameters of different tasks in *MOREC-MOPO*

| TASK | Temperature coefficient $\kappa$ | Penalty coefficient $\beta$ |
|---|---|---|
| HalfCheetah-v3-H | 0 | 2.0 |
| HalfCheetah-v3-L | 0 | 2.0 |
| HalfCheetah-v3-M | 0 | 2.0 |
| Hopper-v3-H | 0 | 20.0 |
| Hopper-v3-L | 0 | 2.0 |
| Hopper-v3-M | 0 | 20.0 |
| Walker2d-v3-H | 0 | 0.1 |
| Walker2d-v3-L | 0 | 2.0 |
| Walker2d-v3-M | 0 | 2.0 |
| halfcheetah-medium-expert-v2 | 0 | 2.5 |
| halfcheetah-medium-replay-v2 | 0.1 | 0.5 |
| halfcheetah-medium-v2 | 0.1 | 0.5 |
| halfcheetah-random-v2 | 0.1 | 0.5 |
| hopper-medium-expert-v2 | 0 | 15.0 |
| hopper-medium-replay-v2 | 0 | 10.0 |
| hopper-medium-v2 | 0.1 | 15.0 |
| hopper-random-v2 | 0 | 10.0 |
| walker2d-medium-expert-v2 | 0 | 1.25 |
| walker2d-medium-replay-v2 | 0 | 0.25 |
| walker2d-medium-v2 | 0.1 | 0.5 |
| walker2d-random-v2 | 0.1 | 1.0 |

Table 7: Particular hyper-parameters of different tasks in *MOREC-MOBILE*

| TASK | Temperature coefficient $\kappa$ | Penalty coefficient $\beta$ |
|---|---|---|
| HalfCheetah-v3-H | 0 | 0.8 |
| HalfCheetah-v3-L | 0 | 0.4 |
| HalfCheetah-v3-M | 0.1 | 0.5 |
| Hopper-v3-H | 0.1 | 2.5 |
| Hopper-v3-L | 0 | 0.4 |
| Hopper-v3-M | 0 | 2.0 |
| Walker2d-v3-H | 0 | 0.04 |
| Walker2d-v3-L | 0 | 0.75 |
| Walker2d-v3-M | 0 | 0.8 |
| halfcheetah-medium-expert-v2 | 0.1 | 1.0 |
| halfcheetah-medium-replay-v2 | 0.1 | 0.2 |
| halfcheetah-medium-v2 | 0.1 | 0.2 |
| halfcheetah-random-v2 | 0 | 0.25 |
| hopper-medium-expert-v2 | 0 | 1.5 |
| hopper-medium-replay-v2 | 0 | 0.6 |
| hopper-medium-v2 | 0.1 | 1.5 |
| hopper-random-v2 | 0.1 | 7.5 |
| walker2d-medium-expert-v2 | 0 | 0.2 |
| walker2d-medium-replay-v2 | 0 | 0.01 |
| walker2d-medium-v2 | 0 | 0.75 |
| walker2d-random-v2 | 0 | 1.0 |

**Model-based offline RL.**

* COMBO(Yu et al., 2021) which applies CQL in dyna-style enforces Q-values small on OOD samples;

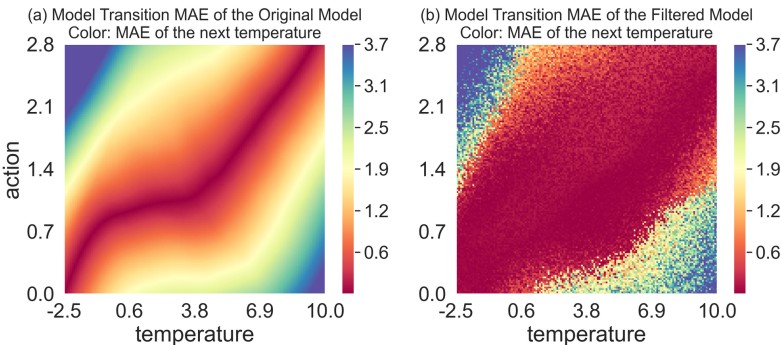

Figure 7: MAE visualizations of the dynamics models with and without transition filtering. The noises of sub-figure (b) come from the sampling process in Eq.(4).

* RAMBO(Rigter et al., 2022) trains the dynamics model adversarially to minimize the value function without loss of accuracy on the transition prediction;
* MOPO (Yu et al., 2020) learns a pessimistic value function from rewards penalized with the uncertainty of the dynamics model's prediction;
* MOBILE (Sun et al., 2023) penalizes the rewards with uncertainty quantified by the inconsistency of Bellman estimations under an ensemble of learned dynamics models.
* TT (Janner et al., 2021) applies a Transformer (Vaswani et al., 2017) to modeling distributions over trajectories and uses beam search for planning.

### E.4   COMPUTATIONAL INFRASTRUCTURE

All experiments were conducted on a workstation outfitted with an Intel Xeon Gold 5218R CPU, 4 NVIDIA RTX 3090 GPUs, and 250GB of RAM, running Ubuntu 20.04.

## F   ADDITIONAL EXPERIMENT RESULTS

### F.1   ADDITIONAL EXPERIMENT RESULTS IN THE SYNTHESIZED TASK

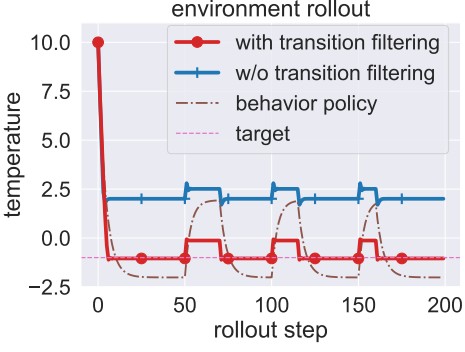

Figure 8: Real environment rollouts of the policies learned with and without transition filtering.

**MAE comparison between the dynamics models with and without transition filtering.** In Fig. 7, as a complementary of Fig. 3 in the main body, we plot the transition error of the original model transition and the model transition filtered by using the dynamics reward. It can be observed that the low error region has been largely expanded, particularly into OOD region. This clearly demonstrates the superior performance of utilizing the dynamics reward for transition filtering.

**Real environment rollout visualizations.** We apply PPO (Schulman et al., 2017) to learn policies from model rollouts generated with and without transition filtering. The rollouts of the learned

policies evaluated in the *true* environment are visualized in Fig. 8. We observe that the policy learned with transition filtering can effectively control the temperature to $-1°C$. Unfortunately, the policy learned without transition filtering induces a temperature that significantly differs from the target. This observation further validate that the dynamics reward can substantially boost the performance of the learned policy.

**Ablation studies on the ensemble technique.** To validate the efficacy of the ensemble technique, we substitute the ensemble discriminator in Eq.(3) by a single discriminator. Particularly, we implement a variant dynamics reward consisting of only the last discriminator. In Table 8, we compare the average model MAE of the original dynamics model, the dynamics model filtered by the ensemble of discriminators, and the dynamics model filtered by a single discriminator. The average model MAE is computed over the full state-action space. It can be observed that the single discriminator performs worse than the ensemble of discriminators. This result implies the ensemble technique indeed contributes to the performance and the generalizability of the dynamics reward.

Table 8: Average model MAE over the full state-action space of the original dynamics model, the dynamics model filtered by the ensemble of discriminators, and the dynamics model filtered by a single discriminator (the last discriminator).

| Model | Average MAE |
|---|---|
| Original Model | 0.226 |
| Model with Transition Filtering (ensemble discriminator) | **0.127** |
| Model with Transition Filtering (single discriminator) | 0.141 |

### F.2 MOREC-MU AND ITS VALIDATIONS

In the refrigerator temperature-control task, we propose a novel strategy that capitalizes on the dynamics reward by integrating it with RL methods. Specifically, we introduce the variant *MOREC-MU* (Model Update), where the model's parameters are iteratively updated through RL to optimize the dynamics reward. This process involves the utilization of PPO (Schulman et al., 2017) to maximize the cumulative dynamics reward. In the MOREC-MU framework, a given policy employs the current model to generate a batch of rollouts. Each transition within these rollouts is then attributed a dynamics reward. Following this, PPO updates the model's parameters to optimize the accumulated dynamics rewards. This procedure is reiterated over multiple epochs to tailor the model to the policy. Notably, we limit the model ensemble size to a single model in MOREC-MU to ensure on-policy sampling.

To ascertain the efficacy of the dynamics reward in enhancing model accuracy concerning a test policy, we updated the model, initially trained by Eq.(1), using MOREC-MU, employing 60 PPO updating epochs. The updated model's rollout is depicted in Fig. 10. It is evident that the rollouts from the MOREC-MU updated model closely align with the environment, underscoring its capability to refine the model for unobserved test policies. Further insights are offered in Fig. 9, where the model's MAE is visualized alongside the test policy. Contrasting with Fig. 2 (c), the adapted model showcases a significant alteration in the MAE map, with the test policy predominantly situated within the low-MAE zones. This implies that the revised model can offer more precise transitions for the test policy.

Table 9: Mean absolute temperature errors of the learned policies.

| | with transition filtering | with model updating | w/o transition filtering | behavior policy |
|---|---|---|---|---|
| **Mean Temperature Error** | 0.39 | **0.37** | 3.20 | 1.35 |
| **Return** | $-78.0$ | **$-74.0$** | $-640.0$ | $-270.0$ |

Lastly, we seamlessly integrate MOREC-MU into the optimal policy learning procedure. We also consider temperature-control task, which aims to stabilize the refrigerator temperature at $1°C$. We adopt an iterative updating strategy comprising two main steps: (i) refining the policy within the current model to maximize the task reward, and (ii) adjusting the model using the present policy to maximize the dynamics reward. This iterative process is performed for a total of $50$ cycles.

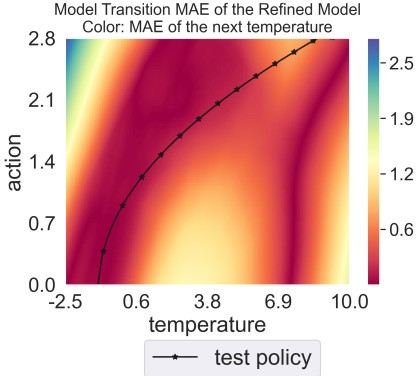

Figure 9: Visualization of the model MAE of the model updated via MOREC-MU. The test policy is surrounded by dark red, corresponding to low MAE.

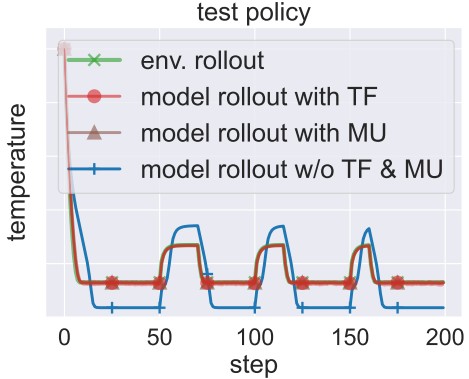

Figure 10: 200-step rollouts in model or environment with the test policy.

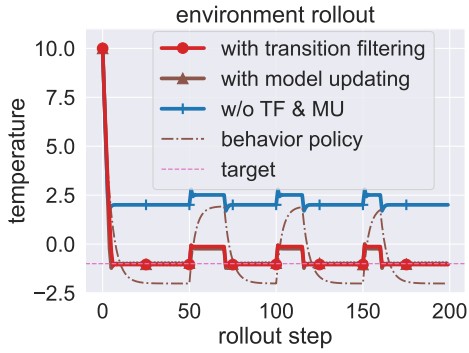

Figure 11: Rollouts of the learned policies in the environment.

Specifically, each iteration involves a two-phase update: a 5-epoch PPO update for the policy, followed by a 5-epoch PPO update for the dynamics model. The resulting rollouts, generated by the refined policy in the environment, are illustrated in Fig. 11. Additionally, we provide a quantitative evaluation of the mean temperature error and return in Table 9. Evidently, MOREC-MU demonstrates superior performance compared to MOREC, particularly evident when regulating the temperature during instances like the refrigerator door being open. This is further corroborated by the results presented in Table 9, where MOREC-MU yields reduced temperature errors.

Collectively, these experimental outcomes highlight an alternative yet promising approach to leveraging the dynamics rewards, showcasing the notable improvement of MOREC-MU over MOREC. These findings not only advocate for the potential merger of MOREC with model updating during policy learning but also emphasize the adaptability of MOREC-MU, which can perpetually refine model parameters in alignment with the prevailing policy, thus producing more precise transitions.

## F.3 MEMORY COST OF THE ENSEMBLE DYNAMICS MODEL

In Algorithm 2, we utilize an ensemble approach to formulate the dynamics reward function. Given the necessity to store and manage historical discriminators, one might anticipate a rising memory overhead with MOREC. Nonetheless, as demonstrated in Table 10, the memory consumption associated with the dynamics reward remains relatively modest. Here, the ensemble size is 40.

Table 10: The memory cost of the dynamics reward in D4RL tasks.

| Task Category Name | Memory Cost (MB) |
|---|---|
| halfcheetah | 36.843 |
| hopper | 36.257 |
| walker2d | 36.843 |

## F.4 TIME COST ANALYSIS

To evaluate the computational impact of MOREC, we compared the runtime of MOREC-MOPO and MOPO with identical hyper-parameters. The mean computational time for each epoch in the `halfcheetah-medium-v2` task is summarized in Table 11. Over a span of 3000 training epochs, MOREC-MOPO exhibits an additional overhead of approximately 1.8 seconds per epoch, translating to an increase of roughly 12.4% in computational time. Consequently, training a MOREC-MOPO policy over 3000 epochs requires an extra 1.5 hours compared to MOPO. Despite this increase, the notable performance enhancement achieved by MOREC-MOPO justifies the additional computational expenditure.

Table 11: The time cost of MOPO and MOREC-MOPO in D4RL halfcheetah tasks.

| Task Name | Time Cost (s/epoch) | |
|---|---|---|
| | MOPO | MOREC-MOPO |
| halfcheetah-medium-v2 | 14.5 | 16.3 |

## F.5 PERFORMANCE OF THE LEARNED DYNAMICS MODEL

In accordance with methodologies established in MOPO (Yu et al., 2020) and MOBILE (Sun et al., 2023), we also train a suite of dynamic models using supervised learning. This is accomplished by optimizing the log-likelihood, as depicted in Eq.(1). Importantly, prior to training, we partition our dataset into training and validation subsets. The model exclusively utilizes the training subset for parameter updates. For reference, the validation root mean square errors (RMSEs) are presented in Table 12.

We also compare the dynamics models obtained through adversarial training (Alg. 2) with those obtained through supervised learning, and the root mean square model errors of part of tasks are listed in Table 13. It can be observed that the dynamics models obtained through adversarial training typically have a higher RMSE error than those obtained through supervised learning. This does not contradict previous research findings on adversarial model learning (Xu et al., 2022a; Shi et al., 2019). In our experiments, we did not fine-tune the hyper-parameters during our adversarial training and only set the rollout horizon to 1. Therefore, the performance of the model obtained through adversarial training is significantly suppressed. Due to the higher model error, we only use the model learned via supervised learning for policy learning (Alg. 1).

## F.6 ADDITIONAL PERFORMANCE RESULTS

We consolidate Table 2 and Table 3 into Table 14. It is evident from the data that MOREC-MOBILE achieves an average score of 80.0, surpassing the prior SOTA results by a margin of 6.7. Notably, MOREC-MOBILE is the sole method reaching the score no less than 80.0. Additionally, it successfully addresses 9 out of 21 tasks, marking an 80% enhancement compared to the previous SOTA.

The learning curves of both MOREC-MOPO and MOREC-MOBILE are depicted in Fig. 12 and Fig. 13. These curves indicate that both methodologies maintain a stable learning progression. Furthermore, the small shaded region, representing the standard error, underscores the robustness of MOREC against variability in random seed initialization.

Table 12: The model root mean square error (RMSE) $\pm$ standard error on the validation data, averaged over 5 seeds.

|  | Model Validation Error |
|---|---|
| HalfCheetah-v3-H | $0.5273 \pm 0.0108$ |
| HalfCheetah-v3-L | $0.4164 \pm 0.0076$ |
| HalfCheetah-v3-M | $0.4917 \pm 0.0095$ |
| Hopper-v3-H | $0.0615 \pm 0.0026$ |
| Hopper-v3-L | $0.0878 \pm 0.0017$ |
| Hopper-v3-M | $0.0664 \pm 0.0010$ |
| Walker2d-v3-H | $0.4785 \pm 0.0069$ |
| Walker2d-v3-L | $0.5046 \pm 0.0052$ |
| Walker2d-v3-M | $0.4488 \pm 0.0066$ |
| halfcheetah-medium-expert-v2 | $0.4281 \pm 0.0185$ |
| halfcheetah-medium-replay-v2 | $0.7182 \pm 0.0077$ |
| halfcheetah-medium-v2 | $0.4742 \pm 0.0071$ |
| halfcheetah-random-v2 | $0.3449 \pm 0.0039$ |
| hopper-medium-expert-v2 | $0.0443 \pm 0.0008$ |
| hopper-medium-replay-v2 | $0.0734 \pm 0.0024$ |
| hopper-medium-v2 | $0.0609 \pm 0.0027$ |
| hopper-random-v2 | $0.0322 \pm 0.0041$ |
| walker2d-medium-expert-v2 | $0.3361 \pm 0.0047$ |
| walker2d-medium-replay-v2 | $0.6256 \pm 0.0076$ |
| walker2d-medium-v2 | $0.5745 \pm 0.0070$ |
| walker2d-random-v2 | $0.5963 \pm 0.0071$ |

Table 13: Root mean square model error comparisons between adversarial-learning models and supervised-learning models.

| Dataset | Adversarial Learning | Supervised Learning |
|---|---|---|
| walker2d-medium-expert-v2 | 0.6321 | **0.3361** |
| walker2d-medium-replay-v2 | 1.3427 | **0.6256** |
| walker2d-medium-v2 | 0.6589 | **0.5745** |
| walker2d-random-v2 | 2.7218 | **0.5963** |
| **Average** | 1.3389 | **0.5331** |

## F.7 ADDITIONAL ABLATION STUDIES

In order to validate the effectiveness of the transition filtering technique, as in Fig. 6, we implement a variant of MOREC, which is not equipped with the transition filtering technique. We set the hyper-parameters of the variant to be the same as MOREC. We then train policies with both methods in the `hfctah` tasks of the D4RL benchmark. The final policy performance is shown in Table 15. We can find the policy performance of MOREC-MOPO dropped a lot without the transition filtering technique, implying the significance of the technique.

Besides, we additionally consider the maximum transition MAE. For each model rollout, we obtain all its transition MAEs and choose the maximum one for this rollout. Then, we take the average of these maximum MAEs across all rollouts collected during full training process and obtain the *maximum transition MAE*. We list the maximum transition MAE for 12 D4RL tasks in Table 16, showing a significantly MAE reduction of MOREC-MOPO. This result implies MOREC-MOPO can avoid inserting the outlier transitions to its replay buffer and ensure the training stability.

## F.8 RESULTS OF ADDITIONAL OFFLINE RL TASKS

In addition to evaluating performance on locomotion tasks within the D4RL and NeoRL benchmarks, our study extends to the Adroit domain, which has been utilized in prior research, such as EDAC (An et al., 2021) and MOBILE (Sun et al., 2023). Specifically, we focus on two tasks: `pen-cloned-v1` and `pen-human-v1`. We employ the MOREC-MOBILE method and present the comparative

Table 14: Performances with maximum improvement ratios in D4RL and NeoRL benchmarks. Averaged over 5 seeds. The previous SOTA algorithm is subscripted.

| TASK | MOREC-MOPO | MOREC-MOBILE | PREV-SOTA | Max Improvement Ratio |
|---|---|---|---|---|
| HalfCheetah-v3-H | 90.27 | **98.50** | 83.00$_{(MOBILE)}$ | 15.50 |
| HalfCheetah-v3-L | 53.50 | 51.22 | **54.70**$_{(MOBILE)}$ | -1.20 |
| HalfCheetah-v3-M | 84.08 | **85.99** | 77.80$_{(MOBILE)}$ | 8.19 |
| Hopper-v3-H | 72.77 | **108.48** | 87.80$_{(MOBILE)}$ | 20.68 |
| Hopper-v3-L | 25.36 | **26.75** | 18.30$_{(MOBILE)}$ | 8.45 |
| Hopper-v3-M | 83.50 | **109.23** | 70.30$_{(TD3+BC)}$ | 38.93 |
| Walker2d-v3-H | **83.04** | 79.30 | 74.90$_{(EDAC)}$ | 8.14 |
| Walker2d-v3-L | **65.01** | 56.81 | 44.70$_{(CQL)}$ | 20.31 |
| Walker2d-v3-M | **76.59** | 71.07 | 62.20$_{(MOBILE)}$ | 14.39 |
| halfcheetah-medium-expert-v2 | **112.07** | 110.94 | 108.20$_{(MOBILE)}$ | 3.87 |
| halfcheetah-medium-replay-v2 | **76.45** | 76.40 | 71.70$_{(MOPO)}$ | 4.75 |
| halfcheetah-medium-v2 | **82.27** | 82.12 | 77.90$_{(MOBILE)}$ | 4.37 |
| halfcheetah-random-v2 | 51.57 | **53.19** | 39.30$_{(MOBILE)}$ | 13.89 |
| hopper-medium-expert-v2 | **113.25** | 111.47 | 112.60$_{(MOBILE)}$ | 0.65 |
| hopper-medium-replay-v2 | 105.15 | **105.45** | 103.90$_{(MOBILE)}$ | 1.55 |
| hopper-medium-v2 | 106.96 | **107.97** | 106.60$_{(MOBILE)}$ | 1.37 |
| hopper-random-v2 | **32.10** | 26.58 | 31.90$_{(MOBILE)}$ | 0.20 |
| walker2d-medium-expert-v2 | **115.77** | 115.52 | 115.20$_{(MOBILE)}$ | 0.57 |
| walker2d-medium-replay-v2 | 95.50 | **95.83** | 89.90$_{(MOBILE)}$ | 5.93 |
| walker2d-medium-v2 | 89.93 | 85.76 | **92.50**$_{(EDAC)}$ | -2.57 |
| walker2d-random-v2 | **23.53** | 22.75 | 16.60$_{(MOBILE)}$ | 6.93 |
| Average | 78.0 | **80.0** | 73.3 | 8.3 |
| Solved tasks (performance > 95) | 6/21 | **9/21** | 5/21 | - |

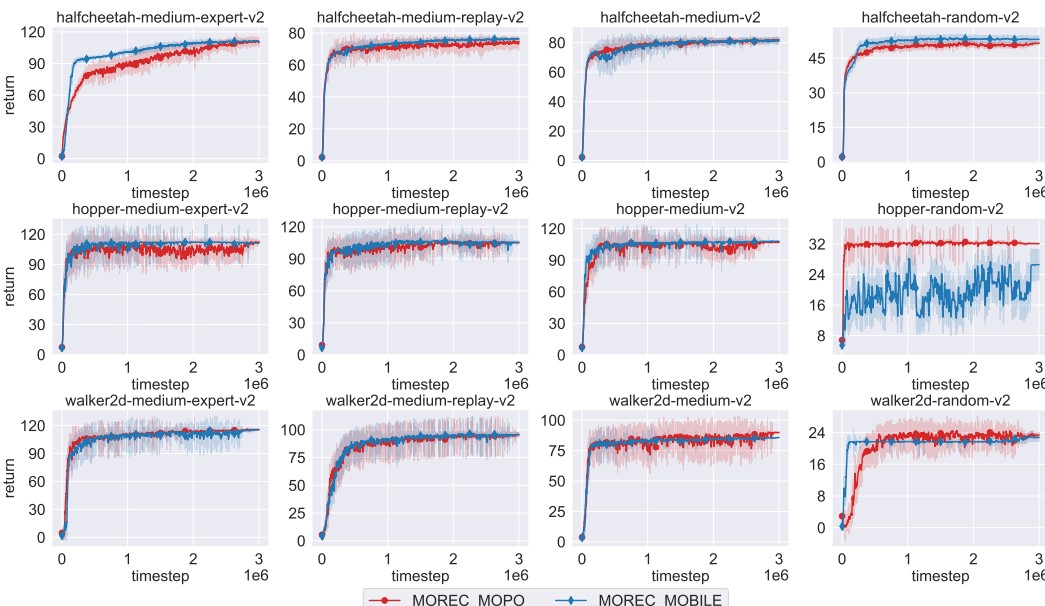

Figure 12: The learning curves of MOREC-MOPO and MOREC-MOBILE in D4RL tasks. The curves are shaded with 1 standard error over 5 seeds.

results in Table 17. The comparative analysis reveals that MOREC-MOBILE achieves superior performance relative to both MOPO and MOBILE across these tasks. However, it is noteworthy that in the `pen-human-v1` task, which is characterized by a smaller offline dataset, MOREC-MOBILE does not perform as well as EDAC. This performance discrepancy is likely attributable to the reduced size of the dataset, which may constrain the model performance more significantly than in the `pen-cloned-v1` task.

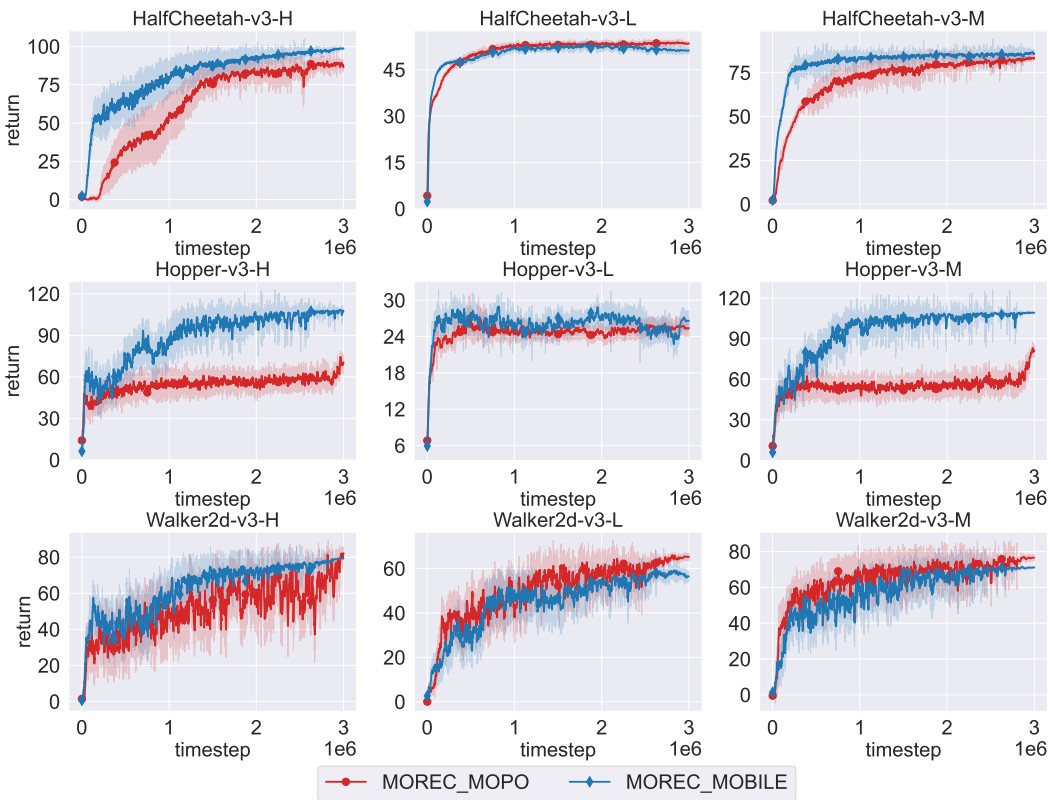

Figure 13: The learning curves of MOREC-MOPO and MOREC-MOBILE in NeoRL tasks. The curves are shaded with one standard error over 5 seeds.

Table 15: The final normalized policy returns of *MOREC-MOPO* and a variant of *MOREC-MOPO* without transition filtering, averaged over 5 seeds.

|  | rnd | med | med-rep | med-exp |
|---|---|---|---|---|
| **MOREC-MOPO** | $51.6 \pm 0.5$ | $82.3 \pm 1.1$ | $76.5 \pm 1.2$ | $112.1 \pm 1.8$ |
| **without transition filtering** | $38.4 \pm 2.1$ | $44.2 \pm 4.5$ | $73.3 \pm 1.5$ | $80.4 \pm 9.2$ |

### F.9 ADDITIONAL DYNAMICS REWARD DIFFERENCES VISUALIZATIONS

We analyze the relationship between dynamics rewards and the model transition MAE, denoted as $\|s_{t+1} - s^{\star}_{t+1}\|_1$, across 12 D4RL tasks as shown in Fig. 14. Utilizing policies derived from behavior cloning, we sample trajectories from each learned dynamics model, without transition filtering. The joint distribution of $\{(\|s_{t+1} - s^{\star}_{t+1}\|, r^D(s_t, a_t, s^{\star}_{t+1}) - r^D(s_t, a_t, s_{t+1}))\}$ is depicted in Fig. 14, with $r^D(\cdot)$ representing the dynamics reward, $s^{\star}_{t+1}$ the true next state, and $s_{t+1}$ the next state predicted by the model. It is noteworthy that trajectories in the random data subset are drawn from a random policy, resulting in a tighter distribution compared to other data sources.

In tasks such as walker2d and hopper, the discrepancies in dynamics rewards are predominantly positive, underscoring the potent discriminative prowess of the dynamics reward. This indicates that, for these tasks, the dynamics reward consistently assigns higher values to true transitions over model transitions. The distribution patterns for walker2d are distinct from those of hopper. This distinction stems from the variation in one-step model errors between these tasks, as elaborated in Table 12. The root mean square error (RMSE) for walker2d is nearly an order of magnitude greater than hopper. Consequently, cumulative errors in hopper are more contained, leading to a smaller $x$-axis range. Specifically, the $x$-axis span for hopper is roughly $[0.001, 0.5]$, in contrast to $[0.008, 7]$ for walker2d. In walker2d, dynamics rewards begin to exhibit significant differences

Table 16: The average maximum transition MAE $\pm$ standard error, averaged over 5 seed, in 12 D4RL tasks. For each model rollout, we obtain all its transition MAE and choose the maximum one for the model rollout. Then, we take the average of these maximum MAEs across all rollouts and obtain the maximum transition MAE.

|  | **MOREC-MOPO** | **without transition filtering** |
|---|---|---|
| halfcheetah-medium-expert-v2 | **6.066 $\pm$ 1.777** | 249906.304 $\pm$ 48324.466 |
| halfcheetah-medium-replay-v2 | **3.860 $\pm$ 0.102** | 11.653 $\pm$ 5.191 |
| halfcheetah-medium-v2 | **3.282 $\pm$ 0.049** | 304736.147 $\pm$ 71894.240 |
| halfcheetah-random-v2 | **3.137 $\pm$ 0.252** | 212908.396 $\pm$ 33037.017 |
| hopper-medium-expert-v2 | **0.556 $\pm$ 0.024** | 1.553 $\pm$ 0.300 |
| hopper-medium-replay-v2 | **0.598 $\pm$ 0.005** | 0.613 $\pm$ 0.014 |
| hopper-medium-v2 | **0.529 $\pm$ 0.004** | 0.581 $\pm$ 0.018 |
| hopper-random-v2 | **0.183 $\pm$ 0.003** | 0.185 $\pm$ 0.004 |
| walker2d-medium-expert-v2 | **3.411 $\pm$ 0.160** | 8.453 $\pm$ 0.660 |
| walker2d-medium-replay-v2 | **5.619 $\pm$ 0.130** | 8.568 $\pm$ 0.570 |
| walker2d-medium-v2 | **3.892 $\pm$ 0.178** | 4.532 $\pm$ 0.360 |
| walker2d-random-v2 | **6.964 $\pm$ 0.601** | 12.640 $\pm$ 1.344 |

Table 17: Normalized average returns in 2 D4RL Adroit tasks, averaged over 5 seeds. The previous SOTA methods are underlined.

|  | BC | CQL | EDAC | MOPO | MOBILE | MOREC-MOBILE |
|---|---|---|---|---|---|---|
| pen-cloned-v1 | 38.3 | 27.2 | 68.2 | 54.6 | 69.0 | **74.3 $\pm$ 14.4** |
| pen-human-v1 | 25.8 | 35.2 | **52.1** | 10.7 | 30.1 | 37.3 $\pm$ 18.8 |

when MAE approaches $0.3$, which aligns with the maximum transition MAE observed in `hopper`. This masks the potential trend of increasing dynamics reward discrepancies with escalating transition MAEs.

As for the `halfcheetah` tasks, on the contrary, the dynamics reward function sometimes gives higher reward to the model transition than the true transition, which is not desired. We believe that this phenomenon is due to the limited generalization range of the dynamics reward. In some `halfcheetah` tasks, when the transition MAE is too large, it exceeds the generalization range of the dynamics reward. The experiment results in Fig. 14 indicate that, in the `halfcheetah` tasks, the generalization range of the dynamics reward is within the scope where the MAE is less than $1.0$. In order to discover why MOREC also works in the `halfcheetah` tasks, we additionally visualize the joint distribution of $\{(\|s_{t+1} - s_{t+1}^{\star}\|, r^D(s_t, a_t, s_{t+1}))\}$ in Fig. 15. We find the dynamics reward of `halfcheetah` shows a similar trend as `walker2d`: As the model transition MAE increase, the dynamics rewards reduce gradually. Because the behavior policy of `halfcheetah-random-v2` is a random policy, its distribution is narrow and cannot show the aforementioned trend. As a result, the dynamics reward can still give low dynamics reward to the high-MAE transitions. Consequently, the rollout can be terminated on time before the MAE diverges.

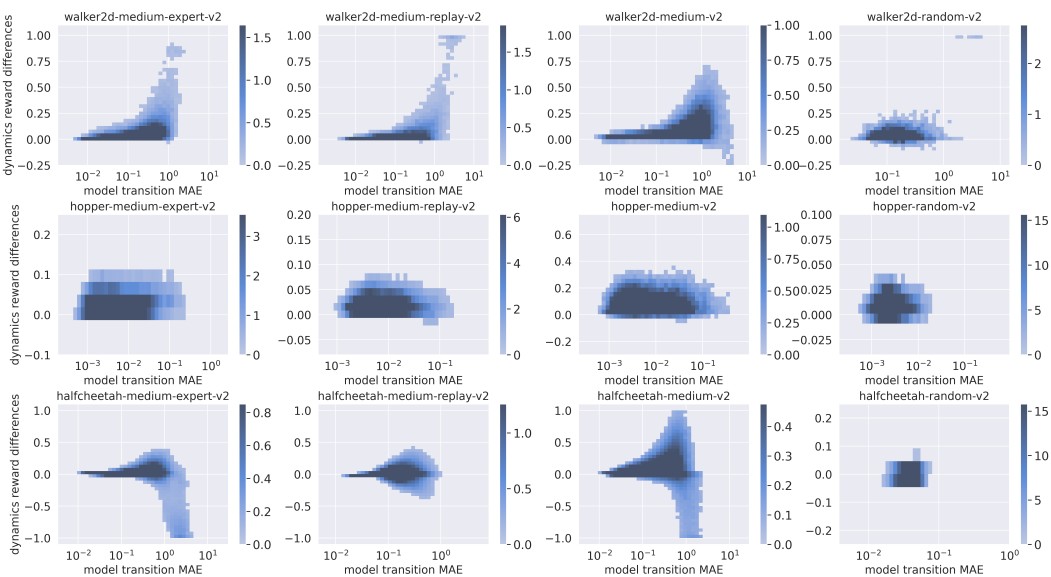

Figure 14: The joint distribution of the dynamics reward differences $r^D(s_t, a_t, s_{t+1}^\star) - r^D(s_t, a_t, s_{t+1})$ and the model transition MAEs $\|s_{t+1} - s_{t+1}^\star\|_1$ in 12 D4RL tasks.

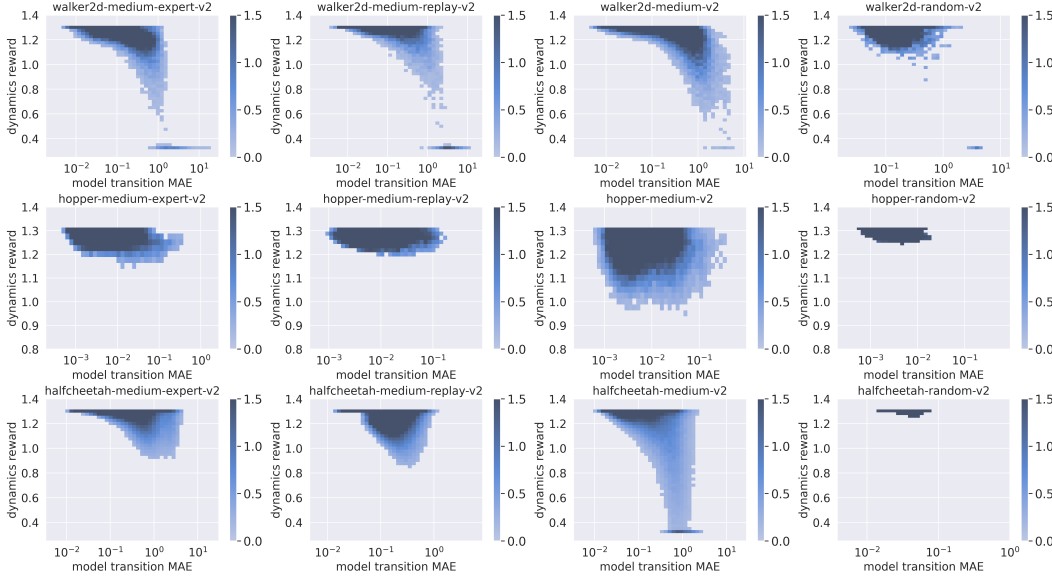

Figure 15: The joint distribution of the dynamics reward for model transitions $r^D(s_t, a_t, s_{t+1})$ and the model transition MAEs $\|s_{t+1} - s_{t+1}^\star\|_1$ in 12 D4RL tasks.

