# OpenReview forum: "Reward-Consistent Dynamics Models are Strongly Generalizable for Offline Reinforcement Learning"
_ICLR.cc/2024/Conference — ICLR 2024 spotlight_

### Official Review · Reviewer_F8oz · 2023-10-27

**Soundness:** 2 fair
**Presentation:** 3 good
**Contribution:** 3 good
**Rating:** 6
**Confidence:** 4

**Summary:**

This paper proposes a method that constructs a discriminator (known as dynamic reward) to distinguish real offline data and generated data from the learned dynamics model. The resulting discriminator acts as a filter for out-of-distribution (or unlikely) transitions. Consequently, the paper claims through experiments that the proposed method achieves out-of-distribution generalization with the dynamic reward, therefore allowing improving existing offline MBRL methods.

**Strengths:**

- The idea is simple and appears to be yield strong results in locomotion tasks.
- The experimental analyses are thorough, in particular:
	- The joint distribution analysis on the learned dynamics rewards
	- The ablation on the filtering

**Weaknesses:**

The following should be addressed:
- For the dynamics reward learning algorithm, why is GAIL better than GAN? It seems like the offline data set is fixed according to Algorithm 1, thus you can treat the dynamics models $P_t$ as the generative models while the dynamics rewards $D_t$ as the discriminators.
- Proposition 1 is a result for tabular setting. Furthermore, this discriminator only differentiates properly the in-distribution transitions---how does this discriminator enable OOD generalization?
	- On page 27, last paragraph indicates that in the halfcheetah task, the dynamics reward appears to be less robust on OOD samples---is this not contradictory to what the main paper claims? Perhaps this has to do with the coverage of the data set---it appears that both `halfcheetah-medium-expert` and `halfcheetah-medium` are visibly suffering in OOD generalization, but they have less coverage on the state-action space compared to `random` and `replay`.
- The rate is $1/\sqrt{T}$, so how big should $T$ really be in practice? Proposition 1 somewhat suggests that $T$ should be almost covering the whole state-action space---seems like in that case we can just directly model the dynamics using a table?
	- Experimentally, does this correspond to 5000 as suggested by Table 4 in the appendix?
- In experimental analysis, it appears the paper focuses mainly on locomotion tasks, and they seem to be mostly overlaps (i.e. NeoRL and D4RL share very similar environments.) I am curious if this applies to other non-trivial environments that are supported by D4RL.

**Questions:**

- Page 2, lines 3-4. It is unclear why the task reward function is known is reasonable given the explanation---it perhaps reinforces that reward function should not be known since it can be considered to be part of the dynamics model?
- Page 2, last third line of first paragraph. What does it mean by "static capabilities"? Does it mean the model is no longer trained after (i) before equation (1)?

**Possible typos**
- Page 1, paragraph 2, line 3: "on the models" instead of "on models"
- Page 1, paragraph 2, line 4, "the model errors" instead of "model errors"
- In Algorithm 1, line 3: Is $\phi$ the same as the policy $\phi$?

---

> ### Author Response · Authors · 2023-11-16
> **Response to Reviewer F8oz-1/2**
>
> We sincerely appreciate the time and effort you have invested in reviewing our paper, as well as your insightful and constructive feedback. Your comments have greatly assisted us in improving the quality of our work.
>
> **Comment 1:** For the dynamics reward learning algorithm, why is GAIL better than GAN? It seems like the offline data set is fixed according to Algorithm 1, thus you can treat the dynamics models as the generative models while the dynamics rewards  as the discriminators.
>
> **Answer C2:** We would like to clarify that we do not claim that GAIL is better than GAN. Actually, they are for different purposes but with the same learning objective. In GAN, the generator inputs a random variable and outputs a data (e.g., image), meanwhile in GAIL the generator is reinforcement learning. Here, learning the dynamics reward involves a reinforcement learning task in general, and we employ GAIL framework. Although GAN can be applied in the specific setting that horizon=1 in our experiment, we believe GAIL is suitable for general situations.
>
> **Comment 2:** Proposition 1 is a result for tabular setting. Furthermore, this discriminator only differentiates properly the in-distribution transitions---how does this discriminator enable OOD generalization?
>
> - On page 27, last paragraph indicates....
>
> **Answer C2:** To address your first concern, we extend Proposition 1 to the linear function approximation setting [1]. In this setting, the dynamics models and discriminators are represented by linear functions with respect to the features. Compared to the tabular result, the difference is that the state-action space size on error terms is replaced by the feature dimension. We will involve this result in the revised paper.
>
> For your second question, unfortunately, we do not have a theoretical answer to explain the OOD generalization ability of the dynamics reward. Nevertheless, we can find that the use of the ensemble of discriminators (in Eq.(3)) can be helpful. Specifically, in the following table, we list the average model MAE of the original dynamics model, the dynamics model filtered by the ensemble of discriminators, and the dynamics model filtered by a single discriminator (the last discriminator). It can be observed that the single discriminator performs worse than the ensemble of discriminators. However, we believe this is not the final answer. We will continue to study the theoretical properties of the dynamics reward in the future.
>
> |                                                       | Average Model MAE |
> | ----------------------------------------------------- | ----------------- |
> | original model                                        | 0.226             |
> | model with transition filter (ensemble discriminator) | 0.127             |
> | model with transition filter (single discriminator)   | 0.141             |
>
> Finally, we would like to point out that the experimental results in Figure 12 on page 27 is not contradictory to our main claim. We do not claim that the dynamics reward is always robust to samples from any level of out-of-distribution. Essentially, the dynamics reward is robust to OOD samples within a region but its robustness also has limit. The results in Figure 12 exactly validate this claim. On one hand, for OOD samples with MAE less than 1.0, the dynamics reward exhibits the remarkable robustness. On the other hand, the dynamics reward cannot show unlimited robustness for extremely OOD samples with MAE around 10.0. From the information-theoretic perspective, we believe that no method can achieve unlimited robustness to samples from any level of OOD. It is a promising direction to further expand the robustness boundary of our dynamics reward learning method.
>
> Reference:
>
> [1] Chi Jin, et al. “Provably eﬃcient reinforcement learning with linear function approximation.” COLT 2020.
>
> **Comment 3:** The rate is $1/\sqrt{T}$, so how big $T$ should really be in practice? Proposition 1 somewhat suggests that should be almost covering the whole state-action space---seems like in that case we can just directly model the dynamics using a table?
>
> - Experimentally, does this correspond to 5000 as suggested by Table 4 in the appendix?
>
> **Answer C3:** We appreciate your question. In practice, we choose the number of iterations $T$ such that the losses of the discriminator and dynamics model have converged. In our experiments, this indeed corresponds to 5000 as shown in Table 4.
>
> In the tabular setting, $T$ should cover the state-action space and the dynamics model is represented by a table. As mentioned in **Answer C1**, we can extend Proposition 1 to the linear function approximation setting. In this setting, the dynamics model is represented by a linear function with respect to the feature, and $T$ does not need to cover the whole state-action space and only depends on the feature dimension.

---

> > ### Author Response · Authors · 2023-11-16
> > **Response to Reviewer F8oz-2/2**
> >
> > **Comment 4:** In experimental analysis, it appears the paper focuses mainly on locomotion tasks, and they seem to be mostly overlaps (i.e. NeoRL and D4RL share very similar environments.) I am curious if this applies to other non-trivial environments that are supported by D4RL
> >
> > **Answer C4:** To address this concern, we run experiments on a more challenging Adroit domain, which is used in existing works like EDAC [An et al., 2021] and MOBILE [Sun et al., 2023]. Due to time limit, we choose two tasks `pen-cloned-v1` and `pen-human-v1`, and run the method MOREC-MOBILE. The results are reported in the following table. We see that MOREC-MOBILE consistently outperforms MOPO and MOBILE. We are still running experiments on other tasks and will provide more results in the future revision.
> >
> > |               | BC     | EDAC            | MOPO   | MOBILE | MOREC-MOBILE             |
> > | ------------- | ------ | --------------- | ------ | ------ | ------------------------ |
> > | pen-cloned-v1 | $38.3$ | $68.2$          | $54.6$ | $69.0$ | $\mathbf{74.3 \pm 14.4}$ |
> > | pen-human-v1  | $25.8$ | $\mathbf{52.1}$ | $10.7$ | $30.1$ | $37.3 \pm 18.8$          |
> >
> > **Question 1:** Page 2, lines 3-4. It is unclear why the task reward function is known is reasonable given the explanation---it perhaps reinforces that reward function should not be known since it can be considered to be part of the dynamics model?
> >
> > **Answer Q1:** We apologize for this misleading claim. The correct claim is that if the task reward function is unknown, we can easily resolve this issue by incorporating it into the dynamics model and learning this augmented dynamics model.
> >
> > **Question 2:** Page 2, last third line of first paragraph. What does it mean by "static capabilities"? Does it mean the model is no longer trained after (i) before equation (1)?
> >
> > **Answer Q2:** Yes, you are right. We will revise this claim to remove any confusion.
> >
> > **Suggestion 1:**  Possible typos. Page 1, paragraph 2, line 3: "on the models" instead of "on models". Page 1, paragraph 2, line 4, "the model errors" instead of "model errors". In Algorithm 1, line 3: Is $\phi$ the same as the policy $\phi$?
> >
> > **Answer S1:** Thanks for your kind suggestions. In line 3 in Algorithm 1, the notion should be the empty set. We have fixed these typos and throughly revised our paper.
> >
> > ---
> >
> > We hope that the above response can address your concerns adequately. We would greatly appreciate it if you could re-evaluate our paper based on the above responses.

---

> ### Comment · Reviewer_F8oz · 2023-11-19
>
> We thank the authors for the detailed response.
>
> Regarding the answers:
> **Comment 1**: Perhaps this is why I don't completely understand why this setting is GAIL but not GAN. Based on Algorithm 2 the objective appears to be sampling from the offline dataset $\mathcal{D}$, which is supposedly under the i.i.d. supervised learning setting, so how is this RL since $\mathcal{D}$ will not be modified based on $D$ nor $P$.
>
> **Comment 2**:
> - Regarding linear MDPs: I agree the referenced paper will likely give you a result that is dependent on $d$, though I would prefer if the paper can specifically mention this.
> - Regarding OOD capability: I believe we agree that the experiment does not indicate that the method generalizes to extremely OOD samples. Consequently, this raises the question of whether the paper has over-claimed the novelty. In the paper title, the phrase "Strongly Generalizable" may be misleading---I was expecting this approach to also deal with extremely OOD samples. I suggest the paper to avoid using the word "strong" as it appears to be an absolute term, but instead using relative terms (e.g. Better, Stronger, More Generalizable, etc.)
>
> **Comment 3**: This raises the question of scalability. If we start considering image-based tasks (e.g. Atari or robotics), then the number of models required for this approach becomes prohibitively expensive as each models can have millions, if not billions of parameters. I believe the paper should indicate (even better if address) this limitation.
>
> Thank you for the other responses, I acknowledge that I have read them. Due to the above questions, I keep my current score.

---

> > ### Author Response · Authors · 2023-11-20
> > **Response to Reviewer F8oz**
> >
> > Thank you very much for your detailed and constructive feedback. Your insightful comments and suggestions have been immensely helpful in improving the quality of our work. In response to your latest comments, we have made the following clarifications.
> >
> > **Comment 1**: GAIL and GAN.
> >
> > **Response 1**: In our set-up, we consider a special case of RL with horizon=1 (i.e., contextual bandit [1]). As mentioned in your comments, this set-up can also be regarded as the supervised learning set-up. Therefore, both GAIL and GAN can be applied in the set-up of this paper, and it is a promising direction to employ GAN in our dynamics reward learning algorithm. Here we choose GAIL because it is also applicable in a general RL set-up with horizon>1. This general RL set-up is often considered in IRL methods but is out of scope of this paper. We have included the above discussion in the revised paper.
> >
> > [1] Sutton, Richard S., and Andrew G. Barto. "Reinforcement learning: An introduction." MIT press, 2018.
> >
> > **Comment 2.1**: The linear MDPs.
> >
> > **Response 2.1:** We have now uploaded the revised version of our paper. We present the linear function approximation version of Proposition 1 in Proposition 2 in Appendix B.3.
> >
> > **Comment 2.2**: The OOD capability.
> >
> > **Response 2.2:** Thank you for your constructive suggestion. We apologize for the misleading phrase in the title. We will change the title to “Reward-Consistent Dynamics Models are Better Generalizable for Offline Reinforcement Learning” in our revised paper. We have also revised our paper thoroughly to prevent any over-claimed statements.
> >
> > **Comment 3:** The scalability.
> >
> > **Response 3:**  Thank you for pointing out this issue. The discriminator ensemble could consume a significant memory, especially in the tasks with large state and action dimensions, potentially limiting the scalability of MOREC. A possible solution is adopting the last-iterate convergence no-regret learning methods [1, 2], which keep only the last discriminator while maintaining optimal convergence guarantees. We will indicate this in the revised paper.
> >
> > [1] Golowich et al. "Tight last-iterate convergence rates for no-regret learning in multi-player games." NeurIPS 2020.
> >
> > [2] Lei et al. "Last iterate convergence in no-regret learning: constrained min-max optimization for convex-concave landscapes." AISTATS 2021.
> >
> > ---
> >
> >
> >
> > We sincerely appreciate your time and expertise in reviewing our paper. Thank you for guiding us towards making it a better work. We are looking forward to your further response.

---

> > > ### Comment · Reviewer_F8oz · 2023-11-21
> > >
> > > Thank you for the response. With this I am happy to increase the score to 6 as some of the limitations and questions remain.

---

> > > > ### Author Response · Authors · 2023-11-21
> > > >
> > > > We are grateful for the improved score and your helpful comments. Your thoughtful review and insights have significantly contributed to the enhancement of our paper. Thank you again for your time and expertise.

---

### Official Review · Reviewer_zEZ1 · 2023-10-31

**Soundness:** 3 good
**Presentation:** 3 good
**Contribution:** 3 good
**Rating:** 8
**Confidence:** 4

**Summary:**

Model-based RL suffers from compounding error as a result of recursive model rollouts. This is especially pronounced in offline model-based RL, where there is no online data to correct model errors. Prior offline MBRL methods typically address this by discouraging the policy from visiting states where the model is uncertain, via some uncertainty quantification. This paper introduces a novel perspective: they learn a "dynamics reward" that evaluates the dynamics consistency of transitions and use it to filter model rollouts to only produce trajectories that are plausible under the dynamics. Concretely, they train the dynamics reward model using inverse max margin IRL. Rollout filtering is performed in a beam-search fashion, where at each rollout step they sample a number of transitions from the model and weight them by their dynamics rewards. Applying this method on top of existing offline MBRL methods results in significant improvements over prior state-of-the-art on D4RL and NeoRL benchmarks. Overall, this paper makes a solid contribution to offline model-based RL by introducing a novel perspective for tackling compounding error and distribution shift.

**Strengths:**

- This paper introduces a novel and credible remedy for a long-standing problem in model-based RL, namely compounding error due to recursive model rollouts resulting in increasingly out-of-distribution inputs. While prior methods mitigate this issue by discouraging the policy from visiting states where the model is uncertain, this paper proposes to filter model rollouts using an adversarially trained "critic" function. This approach can be easily combined with existing offline MBRL methods and lead to significant performance gain.
- The authors provide extensive evaluations of their method on a variety of domains, ranging from analytical toy tasks to standard benchmarks. Their method demonstrates a universal improvement across these tasks.

**Weaknesses:**

- There are limited ablation studies in the paper. This makes it illusive why the method actually works so well, and when it works well. Please refer to Questions for more details.

**Questions:**

- When training the dynamics reward, the (s, a) pair is always sampled from the dataset, whereas the (s') is either from the dataset or from the adversarial dynamics model. This makes the dynamics reward robust to OOD (s') but not (s, a). How does the model generalize to OOD (s, a)? Can you evaluate the dynamics model explicitly on OOD (s, a) pairs and see if the dynamics reward is still higher with valid transitions?
- How does this method compare to a simple filtering technique like selecting the transition with the lowest bellman error?
- When filtering, why construct a softmax distribution instead of directly taking the argmax action?
- How does the reward dynamics relate to energy-based models? It seems quite similar to an energy function on (s, a, s').

Suggestions:
- Table 1 could use a more detailed caption.
- It would be nice to use this for model-based offline policy evaluation [1]. I would imagine the reduction in compounding error to benefit OPE a lot.

References:
1. Michael R. Zhang, Tom Le Paine, Ofir Nachum, Cosmin Paduraru, George Tucker, Ziyu Wang, Mohammad Norouzi. Autoregressive Dynamics Models for Offline Policy Evaluation and Optimization. ICLR 2021.

---

> ### Author Response · Authors · 2023-11-16
> **Response to Reviewer zEZ1-1/2**
>
> We appreciate your time to review and provide positive feedback for our work.
>
> **Question 1:** When training the dynamics reward, the (s, a) pair is always sampled from the dataset, whereas the (s') is either from the dataset or from the adversarial dynamics model. This makes the dynamics reward robust to OOD (s') but not (s, a). How does the model generalize to OOD (s, a)? Can you evaluate the dynamics model explicitly on OOD (s, a) pairs and see if the dynamics reward is still higher with valid transitions
>
> **Answer Q1:**  Thanks for your insightful questions!
>
> **How does the model generalize to OOD (s, a).** Unfortunately, we do not have a theoretical answer to explain the OOD generalization ability of the dynamics reward. Nevertheless, we find that the use of the ensemble of discriminators (in Eq.(3)) can be helpful. Specifically, in the following table, we list the average model MAE of the original dynamics model, the dynamics model filtered by an ensemble of discriminators, and the dynamics model filtered by a single discriminator (the last discriminator). It can be observed that the single discriminator performs worse than the ensemble of discriminators. However, we believe this is not the final answer. We will continue to study the theoretical properties of the dynamics reward in the future.
>
> |                                                       | Average Model MAE |
> | ----------------------------------------------------- | ----------------- |
> | original model                                        | 0.226             |
> | model with transition filter (ensemble discriminator) | 0.127             |
> | model with transition filter (single discriminator)   | 0.141             |
>
> **Explicitly OOD evaluation**. The performance of the dynamics reward in OOD region can be evaluated from Figure 2(d) and (e). The scores in Figure 2(e) are higher than Figure 2(d) in almost all the area, indicating that the dynamics reward gives higher scores on valid transitions in OOD region. Besides, the results of the D4RL task are shown in Figure 5. For (s, a) pairs with an extremely large MAE (1.0), we can view them as OOD pairs since the dynamics model has extremely poor predictions on such pairs. Figure 5(c) clearly shows that the dynamics rewards of true transitions are higher than that of model transitions on these OOD pairs.
>
> Moreover, to evaluate the dynamics model explicitly on OOD region, we plot the figures in Figure 1 in the [anonymized link](https://anonymous.4open.science/r/iclr24-paper-7694-response-materials-C836/) to compare the original dynamics model and the dynamics model filtered by the dynamics reward. It can be observed that the low error region has been largely expanded into the OOD region.
>
> **Question 2:** How does this method compare to a simple filtering technique like selecting the transition with the lowest bellman error?
>
> **Answer Q2:** Thanks for raising this interesting point. The bellman error is affected by many factors. Firstly, the error of Q function significantly affects the bellman error. How to learn an accurate Q function is still an open problem. Even if we assume that an accurate Q function is available (in this case offline RL has been solved), the bellman error is still affected by the scale of reward and Q values at state-action pairs. Therefore, bellman error is not a proper metric for filtering transitions.
>
> We have conducted comparison experiments on the `halfcheetah` task. For a transition $(s, a, r, s^\prime)$, we calculate the squared Bellman error of $(Q_{\psi} (s, a) - r - \gamma Q_{\psi} (s^\prime, a^\prime))^2$, where $Q_{\psi}$ is the current Q-function, $a^\prime \sim \pi_{\phi} (\cdot|s^\prime)$ and $\pi_{\phi}$ is the current policy. The results are shown in the following table. We observe that the performance of this baseline is significantly worse than MOREC. This demonstrates that the bellman error is not a good metric for transition filtering.
>
> |                     | MOREC-MOPO               | MOPO WITH BELLMAN ERROR FILTER |
> | ------------------- | ------------------------ | ------------------------------ |
> | halfcheetah-med-exp | $\mathbf{112.1 \pm 1.8}$ | $59.1 \pm 9.6$                 |
> | halfcheetah-med-rep | $\mathbf{76.5 \pm 1.2}$  | $57.7 \pm 2.5$                 |
> | halfcheetah-med     | $\mathbf{82.3 \pm 1.1}$  | $67.6 \pm 2.7$                 |
> | halfcheetah-rnd     | $\mathbf{51.6 \pm 0.5}$  | $41.3 \pm 2.18$                |
> | average             | $\mathbf{80.6}$          | $56.4$                         |

---

> > ### Author Response · Authors · 2023-11-16
> > **Response to Reviewer zEZ1-2/2**
> >
> > **Question 3:** When filtering, why construct a softmax distribution instead of directly taking the argmax action?
> >
> > **Answer Q3:** We appreciate your insightful question. In some tasks such as `halfcheetah`, we observe that incorporating more uncertainty into the model rollout process could help improve the generalization performance of the generated rollouts. A similar observation have been presented in the existing MBRL work [1]. Therefore, we choose to construct a softmax distribution.
> >
> > To empirically verify this point, we conducted additional ablation experiments. In the `halfcheetah` tasks, based on MOREC-MOBILE, we employed two strategies for transition filtering: argmax and softmax. The results are listed in the following table. The results indicate that the average performance of the softmax strategy surpasses that of argmax.
> >
> > |                     | SOFTMAX                  | ARGMAX                  |
> > | ------------------- | ------------------------ | ----------------------- |
> > | halfcheetah-med-exp | $\mathbf{110.9 \pm 0.6}$ | $110.8 \pm 1.3$         |
> > | halfcheetah-med-rep | $\mathbf{76.4 \pm 0.6}$  | $68.9 \pm 2.9$          |
> > | halfcheetah-med     | $\mathbf{82.1 \pm 1.2}$  | $77.1 \pm 3.0$          |
> > | halfcheetah-rnd     | $51.7 \pm 1.3$           | $\mathbf{53.2 \pm 1.4}$ |
> > | average             | $\mathbf{80.3}$          | 77.5                    |
> >
> > Reference:
> >
> > [1] Kurtland Chua, et al. “Deep reinforcement learning in a handful of trials using probabilistic dynamics models.” NeurIPS 2018.
> >
> > **Question 4:** How does the reward dynamics relate to energy-based models? It seems quite similar to an energy function on (s, a, s').
> >
> > **Answer Q4:** The dynamics reward model has no connection to energy-based models which represent the density of the offline dataset. It evaluates a transition pair according to its faithfulness, disregard of its density in the data.
> >
> > **Suggestion 1:** Table 1 could use a more detailed caption.
> >
> > **Answer S1:** Thanks for your kind suggestion. We will revised this part in the paper.
> >
> > **Suggestion 2:** It would be nice to use this for model-based offline policy evaluation. I would imagine the reduction in compounding error to benefit OPE a lot.
> >
> > **Answer S2:** Thanks for bringing us this promising future direction. We will explore the potential of applying our method to model-based offline policy evaluation.
> >
> > ---
> >
> > Thank you for taking the time to provide us with your feedback. We appreciate your valuable comments and suggestions, which will help us improve our work. We look forward to receiving your further feedback.

---

> > > ### Comment · Reviewer_zEZ1 · 2023-11-20
> > >
> > > I appreciate the detailed response. It seems that despite the empirical evidence, the reason why dynamics reward is able to generalize to OOD transitions remains elusive. That said, this paper proposes a strong empirical method and opens up an interesting direction for future research. Therefore, I keep my current evaluation.
> > >
> > > One thing to clarify is that by "energy-based model" I don't mean the energy with respect to the dataset distribution, but with respect to the ground truth dynamics distribution. In other words, (s, a, s') that is likely under the truth dynamics has a low energy, whereas (s, a, s') that is unlikely under the true dynamics has a high energy. It would be interesting to investigate the relationship between dynamics rewards and energy functions in future work.

---

> > > > ### Author Response · Authors · 2023-11-20
> > > > **Response to Reviewer zEZ1**
> > > >
> > > > Thank you very much for your thoughtful and constructive feedback. We greatly appreciate your acknowledgement of the potential our paper holds. We also believe that investigating the relationship between dynamics rewards and energy functions presents an interesting and promising avenue for future work.

---

### Official Review · Reviewer_Gxsz · 2023-11-01

**Soundness:** 3 good
**Presentation:** 2 fair
**Contribution:** 3 good
**Rating:** 6
**Confidence:** 3

**Summary:**

This paper presents an offline model-based RL method that aims to reduce the prediction error of dynamics model in OOD states. The main idea is to learn a dynamics reward which is assigned a higher value when the input transition is close to ground-truth transition. This reward is utilized when predicting the future states with dynamics models, by weighting the future predicted states with their predicted dynamics rewards. Experiments first provide a supporting experiment and analysis on synthetic task and then evaluate the proposed method in D4RL and NeoRL offline RL benchmarks.

**Strengths:**

- The idea of learning dynamics reward is new to my knowledge.
- The paper presents helpful analysis and supporting experiments to understand how the proposed method works.
- The proposed method can consistently improve the performance of offline RL methods in most considered tasks.

**Weaknesses:**

- The interpretation of dynamics model as an agent and introducing the concept of reward adds unnecessary complexity to the method and makes it a bit difficult to easily understand the method. Simply formulating the main idea with adversarial generative training and using the score D as a reward could make the paper be more simple and easy to understand.
- The paper introduces multiple hyperparameters and did quite extensive hyperparameters search (e.g., temperature, penalty, and threshold, ..). Making sure that the baseline is fully tuned with the similar resource given to the proposed method could be important for a fair comparison.
- Figure 2 is very difficult to parse and how this leads to the conclusion that dynamics rewards is superior to dynamics model is not clear.
- The additional complexity of learning dynamics reward and implementation complexity might limit the wide adoption of the proposed method.

**Questions:**

- Could the authors answer & address my concerns & questions in Weaknesses?
- What does ``Accordingly, the behavior policy $\pi_{\beta}$ is regarded as the “dynamics model”` mean? It's difficult to parse and seems out-of-context as the behavior policy is not even defined yet before this line.

---

> ### Author Response · Authors · 2023-11-16
> **Response to Reviewer Gxsz-1/2**
>
> Thank you for taking the time to review our paper and providing us with your valuable feedback.
>
> **Comment 1:** The interpretation of dynamics model as an agent and introducing the concept of reward adds unnecessary complexity to the method and makes it a bit difficult to easily understand the method. Simply formulating the main idea with adversarial generative training and using the score D as a reward could make the paper be more simple and easy to understand.
>
> **Answer C1:** We would like to clarify that the interpretation of dynamics model as an agent has been proposed in several previous studies, including [1, 2, 3, 4]. Such perspective is widely recognized and can help understand the motivation of learning the dynamics reward by IRL. Meanwhile, the single score $D$ cannot be simply used as the dynamics reward as it is coupled with the (learning) transition behaviors. We use an ensemble of $D$, which is also motivated from the perspective of treating dynamics model as an agent. We will clarify this in the revised paper.
>
> References:
>
> [1] Venkatraman Arun, et al. ”Improving multistep prediction of learned time series models.” AAAI 2015.
>
> [2] Yueh-Hua Wu, et al. “Model Imitation for Model-Based Reinforcement Learning.” ArXiv: 1909.11821.
>
> [3] Tian Xu, et al. “Error bounds of imitating policies and environments.” NeurIPS 2020.
>
> [4] Zifan Wu, et al. “Models as Agents: Optimizing Multi-Step Predictions of Interactive Local Models in Model-Based Multi-Agent Reinforcement Learning.” AAAI 2023.
>
> **Comment 2:** The paper introduces multiple hyperparameters and did quite extensive hyperparameters search (e.g., temperature, penalty, and threshold, ..). Making sure that the baseline is fully tuned with the similar resource given to the proposed method could be important for a fair comparison.
>
> **Answer C2:** Thanks for your suggestion. We would like to point that we did not use significant additional resources for hyperparameters search compared to the main baselines MOPO and MOBILE. Here we list the hyperparameter search spaces of MOPO, MOBILE and MOREC-MOPO/-MOBILE on D4RL tasks.
>
> For MOPO and MOBILE, we use the implementation in [Sun et al., 2023], which reports the hyperparameter search strategy in Appendix C.2 and C.3 in [Sun et al., 2023].
>
> 1. For MOPO, [Sun et al., 2023] performs a grid search on three parameters: penalty coefficient $\beta$ with 3 choices, rollout horizon $h$ with 2 choices and uncertainty quantifiers with 3 choices. **Therefore, the search space size in MOPO is 18.**
> 2. For MOREC-MOPO, it introduces two additional hyperparameters: the temperature $\kappa$ and threshold $r^D_{\min}$ in transition filtering. The hyperparameter search strategy is detailed in Appendix D.2. However, we did not tune the hyperparameters $r^D_{\min}$ and $h$. For $r^{D}_{\min}$, based on the result shown in Figure 5(a), we choose a fixed value of 0.6 across all tasks. For $h$, we choose a fixed value of 100 across all tasks. Thus we only take a grid search on two parameters: penalty coefficient $\beta$ with 6 choices, temperature $\kappa$ with 2 choices. **The total search space size in MOREC-MOPO is 12.**
> 3. For MOBILE, [Sun et al., 2023] performs a grid search on two parameters: penalty coefficient $\beta$ with 4 choices, rollout horizon $h$ with 2 choices. **Therefore, the search space size in MOBILE is 8.**
> 4. For MOREC-MOBILE, its hyperparameters and search strategy are the same as MOREC-MOPO. **Thus, the search space size in MOREC-MOBILE is 12.**
>
> In summary, MOPO and MOBILE are tuned with comparable resources to MOREC-MOPO/MOBILE for a fair comparison.

---

> > ### Author Response · Authors · 2023-11-16
> > **Response to Reviewer Gxsz-2/2**
> >
> > **Comment 3:** Figure 2 is very difficult to parse and how this leads to the conclusion that dynamics rewards is superior to dynamics model is not clear.
> >
> > **Answer C3:** Thank you for pointing out this issue. We will revise the paper to make Figure 2 easier to understand.
> >
> > To understand why the dynamics reward is helpful, we can compare Figure 2(d) and (e). The dynamics rewards of the true transitions (e) are higher than the rewards of the model (d), in almost all of the region. This shows that the dynamics reward model can serve as a scoring function to evaluate the truthfulness of transitions, even in the OOD region.
> >
> > More straightforwardly, in Figure 1 in the [anonymized link](https://anonymous.4open.science/r/iclr24-paper-7694-response-materials-C836/), we plot the transition error of the original model transition and the model transition filtered by using the dynamics reward. It can be observed that the low error region has been largely expanded, particularly into OOD region. This clearly demonstrates the superior performance of utilizing the dynamics reward for transition filtering.
> >
> >
> > **Comment 4:** The additional complexity of learning dynamics reward and implementation complexity might limit the wide adoption of the proposed method.
> >
> > **Answer C4:** Thanks for pointing out this issue. Here we record the time and memory complexities of MOPO and MOREC-MOPO in the `walker2d-medium-replay-v2` task. The memory and running time of MOREC-MOPO is around 1.5 times of MOPO. We believe that such a computation complexity is acceptable based on the notable performance boost achieved by MOREC. For the implementation complexity, we will open-source the code upon acceptance to facilitate the wide application of our method.
> >
> > |            | RAM (MB) | TOTAL TRAINING TIME COST (HOUR)                              |
> > | ---------- | -------- | ------------------------------------------------------------ |
> > | MOPO       | 3132     | 11.6 (model learning + policy training)                      |
> > | MOREC-MOPO | 4715     | 17.7 (model learning + dynamics reward learning + policy training) |
> >
> > **Question 1:** What does `Accordingly, the behavior policy is regarded as the “dynamics model”` mean? It's difficult to parse and seems out-of-context as the behavior policy is not even defined yet before this line.
> >
> > **Answer Q1:** We are sorry for this misleading argument. The behavior policy $\pi^{\beta}$ is defined as the policy used to collect the offline dataset $\mathcal{D}$. Here we mean that when treating the environment transition as an agent, the behavior policy plays the role of the dynamics model in the view of this agent. We will clarify this point in the revised paper.
> >
> > ---
> >
> > Thanks for your insightful comments. We hope that the above answers can address your concerns satisfactorily and improve the clarity of our main idea. We would greatly appreciate it if you could re-evaluate our paper based on the above responses.

---

> > > ### Comment · Reviewer_Gxsz · 2023-11-17
> > >
> > > Thank you for the response. Could you explain what `Meanwhile, the single score cannot be simply used as the dynamics reward as it is coupled with the (learning) transition behaviors.` means in a bit more detail?

---

> > > > ### Author Response · Authors · 2023-11-17
> > > >
> > > > Thanks for your prompt response! Here we elaborate on this statement. For any iteration $t \in [T]$, the single score $D_t$ is to distinguish the samples generated from $P^\star$ and $P_{t-1}$, and thus couples with the learning transition behavior of $P_{t-1}$. However, an ideal dynamics reward should assign high scores on true transitions and low scores on **any** other transitions. Therefore, the dynamics reward should not be coupled with a specific (false) transition behavior. In order to satisfy this property, we utilize an ensemble of discriminators to construct the dynamics reward. Please let us know if you have further concerns.

---

> > > > > ### Author Response · Authors · 2023-11-20
> > > > > **We have thoroughly revised our paper**
> > > > >
> > > > > Dear reviewer Gxsz, we are deeply grateful for your swift feedback. We have thoroughly revised our paper, incorporating your suggestions with great care. If there are any further concerns or if you need more clarity on any aspect,  please do not hesitate to contact us. We are also more than happy to answer your further questions.

---

> > > > > > ### Comment · Reviewer_Gxsz · 2023-11-21
> > > > > >
> > > > > > Thank you for your response. Some parts got much more clear after the rebuttal phase and I don't have major concerns, so I will update my score to be 6. One last question is, have you tried investigating how good model P learned in Eq 2 is, for instance, in terms of accuracy or downstream performance? I'm asking because you're abandoning it in contrast to other adversarial learning frameworks that aim to learn a good generator by this objective.

---

> > > > > > > ### Author Response · Authors · 2023-11-22
> > > > > > >
> > > > > > > We are pleased to hear that our responses have resolved your concerns!
> > > > > > >
> > > > > > > To answer your question, we evaluated the accuracy of the dynamics model obtained in adversarial learning. We choose the Walker2d task and report the root mean squared error (RMSE) evaluated on the offline dataset. The results show that the dynamics model obtained by adversarial learning has a notably larger RMSE compared to the model obtained by supervised learning. This is possibly because adversarial learning may bring additional training difficulty compared to supervised learning. Thus we only utilize the dynamics reward and abandon the dynamics model.
> > > > > > >
> > > > > > > |  | adversarial learning | supervised learning |
> > > > > > > | --- | --- | --- |
> > > > > > > | walker2d-medium-expert-v2 | 0.6321 | 0.3361 |
> > > > > > > | walker2d-medium-replay-v2 | 1.3427 | 0.6256 |
> > > > > > > | walker2d-medium-v2 | 0.6589 | 0.5745 |
> > > > > > > | walker2d-random-v2 | 2.7218 | 0.5963 |
> > > > > > > | average | 1.3389 | 0.5331 |

---

> > > > > > > > ### Comment · Reviewer_Gxsz · 2023-11-23
> > > > > > > >
> > > > > > > > Thank you, it could be nice to mention this somewhere in the draft and potentially discuss this.

---

> > ### Comment · Reviewer_HqU2 · 2023-11-16
> > **Response to authors**
> >
> > Thank you for the explanations!

---

### Official Review · Reviewer_HqU2 · 2023-11-01

**Soundness:** 4 excellent
**Presentation:** 4 excellent
**Contribution:** 3 good
**Rating:** 6
**Confidence:** 3

**Summary:**

Paper describes a new method for offline model-based RL using a new concept of "dynamics reward", and present results showing its effectiveness.

**Strengths:**

Quality
- Paper presents a number of experiments showing the effectiveness of the proposed method, including SOTA results on standard offline RL benchmarks

Clarity
- Paper was written clearly, and method and experiment were easy to understand

Significance
- Based on the results, I believe this paper will be a nice contribution to the area of offline model-based RL

**Weaknesses:**

- A major contribution of this paper is the new concept of the "dynamics reward", and the algorithm used to learn the dynamics reward model (GAIL). However, it was not clear to me what this dynamics reward should represent. It would be helpful to describe the optimal closed-form solution (if there was one), or include some additional discussion on an intuitive interpretation of it.

**Questions:**

- It seems to me the dynamics reward model kind of serves as a density model of the training data? If it is, then why not just learn a density model? If not, then what is the difference?

---

> ### Author Response · Authors · 2023-11-16
> **Response to Reviewer HqU2**
>
> Thank you for taking the time to review our paper, and for your insightful comments.
>
> **Comment 1:** A major contribution of this paper is the new concept of the "dynamics reward", and the algorithm used to learn the dynamics reward model (GAIL). However, it was not clear to me what this dynamics reward should represent. It would be helpful to describe the optimal closed-form solution (if there was one), or include some additional discussion on an intuitive interpretation of it.
>
> **Answer C1:** Thanks for your valuable comment. We have no optimal closed-form solution due to the minimax formulation. But it can be intuitive to understand the dynamics reward as a scoring function for transitions. An ideal dynamics reward assigns higher scores for real transitions than any other ones. That is, given any state-action pair $(s,a)$ and the real next state $s^\star$, $r^D(s,a,s^\star) > r^D(s,a,s')$ for any unreal next state $s'$. Once such dynamics reward is obtained, it is straightforward to see that the transition function can be faithfully recovered by maximizing the dynamics reward, which is the motivation of this paper.  We will involve the above discussion into the revised paper.
>
> **Comment 2:** It seems to me the dynamics reward model kind of serves as a density model of the training data? If it is, then why not just learn a density model? If not, then what is the difference?
>
> **Answer C2:** From **Answer C1**, the dynamics reward model has no connection to density. It evaluates a transition pair according to its faithfulness, disregard of its density in the data.
>
> We notice that a reader may misunderstand the dynamics reward as a density model from Figure 2(d). In this figure, the high scoring region is close to the high density region. This observation is due to that supervised model learning makes less errors in the high density region, and coincidentally the dynamics reward correctly recognizes the errors of the model. It can be found from Figure 2(e) that, when given real transitions, the high scoring region is dissimilar to the high density region. We will include the above discussion into the revised paper.
>
> ---
>
> Thanks for your valuable feedback. We hope that the above answers can address your concerns satisfactorily and improve the clarity of our major contribution. We are looking forward to your further response.

---

### Author Response · Authors · 2023-11-16
**General Response**

We thank all reviewers for their expertise and efforts in reviewing our paper. We have responded to each review seperately. We hope that our response can address the concerns well. Furthermore, we look forward to any additional comments or suggestions for improvement.

Best,

The Authors

---

### Author Response · Authors · 2023-11-19
**Request for Further Review Feedback**

Dear Reviewers,

We sincerely appreciate the time and effort you have invested in reviewing our paper. We have carefully considered all your feedback and have endeavored to address each concern in our submitted response. As the discussion period is nearing its conclusion, we kindly request any further feedback you may have. Your insights are crucial in enhancing the quality of our work.

If you find that our responses have adequately addressed your concerns, we would be grateful for your consideration in possibly raising the rating of our work. On the other hand, should you have any additional questions or lingering concerns, please do not hesitate to contact us. We are more than willing to provide further clarifications and answer any additional questions you might have.

Once again, we extend our deepest gratitude for your valuable contributions to the review process.

Best Regards,

The Authors

---

### Author Response · Authors · 2023-11-20
**The revised paper has been uploaded**

Dear Reviewers,

We have completed the revisions of our manuscript in response to your insightful feedback. The key changes have been highlighted in blue for ease of identification. Below is a summary of the significant revisions:

**Main text**

- `[HqU2]` (Section 1) Added an intuitive description of the dynamics reward.
- `[F8oz]`  (Section 3.2) Discussed on the application of GAN in dynamics reward learning.
- `[F8oz]`  (Section 3.2) Introduced the linear function approximation version of Proposition 1.
- `[HqU2]` (Section 4.2) Clarified that the dynamics reward is distinct from a density model.
- `[Gxsz]` (Section 4.2) Revised the discussion of the results depicted in Figure 2.
- `[Gxsz]` (Section 4.2) Included the MAE of the dynamics model filtered by the dynamics reward.
- `[F8oz]` (Section 5) Included the discussion on the scalability limitation and potential future direction.

**Appendices**

- `[F8oz]` (Appendix B.3) Added a linear function approximation version of Proposition 1.
- `[zEZ1, F8oz]` (Appendix E.1) Presented the model MAE with and without the ensemble technique.
- `[F8oz]` (Appendix E.8) Presented comparative results on the `Adroit` tasks.
- `[F8oz]` (Appendix E.9) Discussed the tasks where the dynamics reward shows less robustness to OOD samples.

Best,

The Authors

---

### Meta-Review · Area_Chair_Vs6e · 2024-01-10

**Metareview:**

This paper introduces the use of "dynamics reward" to improve Model-based Offline Reinforcement Learning.

This is a novel and very interesting contribution. Although the experiments do not fully shed light into the mechanisms of why dynamic reward improve performances, the empirical evaluation show significant improvements in performance.

All the reviewers agree that the paper is worth publishing, and the authors significantly improved the manuscript during the rebuttal phase.

**Justification For Why Not Higher Score:**

The contribution of this paper is of interest only for a narrow part of the community.

**Justification For Why Not Lower Score:**

All the reviewers agree that the paper is worth publishing. The contribution is novel and empirically important.

---

### Decision · Program_Chairs · 2024-01-16

Accept (spotlight)